# Technical note: Discrete in situ vapor sampling for subsequent lab-based water stable isotope analysis

*Barbara Herbstritt[1], Benjamin Gralher[1,2], Stefan Seeger[1], Michael Rinderer[1,3], Markus Weiler[1]*

[1] *Chair of Hydrology, Faculty of Environment and Natural Resources, University of Freiburg, Germany*
[2] *Institute of Groundwater Management, Technical University of Dresden, Germany*
[3] *now at: geo7 AG, Bern, Switzerland*

## Abstract

Methodological advancements have been made in in situ observations of water stable isotopes that have provided valuable insights in ecohydrological processes. The continuous measurement capabilities of laser-based analyzers allow for high temporal resolutions and non-destructive, minimally invasive study designs of such in situ approaches. However, isotope analyzers are expensive, heavy, and require shelter and access to electrical power which impedes many in situ assays. Therefore, we developed a new, inexpensive technique to collect discrete water vapor samples in the field via diffusion-tight inflatable bags that can later be analysed in the lab. In a series of structured experiments, we tested different procedural settings, bag materials, and closure types for diffusion-tightness during storage as well as for practical handling during filling and extraction. To facilitate re-usage of sampling bags, we present a conditioning procedure using ambient air as primer. In order to validate our method, direct measurements through hydrophobic in situ probes were compared to repeated measurements of vapor sampled with our bags from the same source. All steps are summarized in a detailed SOP. This procedure represents the preparation and measurement of calibration and validation vapor standards necessary for processing of unknown, field-collected vapor samples in the foreseen application. Performing pertinent calibration procedures, accuracy was better than 0.4‰ for $\delta^{18}O$ and 1.9‰ for $\delta^{2}H$ after one day of storage. Our technique is particularly suitable in combination with minimal-invasive water

vapor sampling in situ probes that have already been employed for soils and tree xylem. It is an important step towards minimally invasive monitoring of stable isotope distributions and also time-series in virtually undisturbed soils and trees without the need to have field-access to an analyzer. It is therefore a promising tool for many applications in eco-hydrology and meteorology.

## 1 Introduction

Analyses of stable isotope composition of hydrogen and oxygen ($\delta^2$H and $\delta^{18}$O) in soils and plant water have proven to be powerful tools and are therefore widely employed in ecology, hydrology, and related disciplines. Stable isotopes of pore water have been used to provide insights into soil evaporation (Zimmermann et al., 1967; Allison, 1982; Allison et al., 1983; Barnes and Allison, 1988; Walker et al., 1988) and groundwater recharge rates (Dincer et al., 1974; Saxena, 1984; Darling and Bath, 1988, Koeniger et al., 2016). They were used in soil hydrology to study unsaturated and saturated subsurface flow processes, mixing and residence times (Sklash and Farvolden, 1979; Buttle and Sami, 1990; McDonnell, 1990; Stewart and McDonnell, 1991; Gazis and Feng, 2004; Laudon et al., 2004; Garvelmann et al., 2012; Beyer et al., 2016), and to quantify evapotranspiration partitioning (Brunel et al., 1997; Hsieh et al., 1998a; Yepez et al., 2005; Rothfuss et al., 2010; Wang et al., 2012; Dubbert et al., 2013; Quade et al., 2019). Applications of water stable isotopes in ecology have allowed researchers to identify plant water sources (Dawson & Ehleringer, 1991), to describe water use patterns (Schwinning et al., 2002), and to determine competitive interactions (Ehleringer et al., 1991, Meißner et al., 2012). In plant physiology, insights into plant hydraulic architecture (Drake & Franks, 2003) were possible with isotope techniques, root water uptake was quantified (Rothfuss & Javaux, 2017; Fan et al., 2017; Seeger & Weiler, 2021) as well as hydraulic lift (Caldwell & Richards, 1989; Meunier et al., 2018).

Conventionally, measurements of pore water and tree xylem water isotope composition are obtained through destructive sampling of soil cores or manual collection of sapwood and subsequent water extraction for isotope ratio mass spectrometry (IRMS) analysis (Ehleringer et al., 2000, West et al., 2006) or Isotope Ratio Infrared Spectrometry (IRIS) (Baer et al.,

2002; Gupta et al., 2009). These instruments allow for high measurement precision (Horita & Kendall, 2004), but are comparably expensive in the case of IRMS and generally require highly time-consuming and laborious sample pre-treatment (Kerstel & Gianfrani, 2008). A less expensive and overall more convenient approach relying on laser-based water stable isotope analyzers is the direct vapor equilibration laser spectrometry (DVE-LS) where

samples of soil matrix, rocks or plant tissue are in equilibrium with a corresponding vapor phase (Wassenaar et al., 2008; Hendry et al., 2015; Gralher et al., 2021).

Although promising, the disadvantage is still that destructive soil sampling or harvesting of plant material generally prevents repeated samples from the exact same position. Additionally, repeated sampling of xylem imposes the risk of killing the tree or weakening it

due to fungal infestation. Moreover, taking branch samples can be challenging or even impossible for tall trees. Generally, destructive sampling restricts the number of samples that can be obtained over time and space. This makes high-frequency or even continuous measurements difficult to sustain or simply infeasible. Number and spatiotemporal scope of lab-scale experimental setups as well as environmental isotope studies continued to expand,

but were still limited by the available indirect observational techniques (West et al., 2010).

The growing distribution of laser-based water stable isotope analyzers in recent years also enabled minimal-invasive, direct, continuous, and simultaneous measurements of $\delta^2H$ and $\delta^{18}O$ of water vapor. Only herewith, time series observations from the same point became possible. Available IRIS instruments allow for measurements at a precision and accuracy

comparable to that of IRMS (Berden et al., 2000; Baer et al., 2002; Crosson, 2008; Kerstel

and Gianfrani, 2008; Gupta et al., 2009). Importantly, laser-based instruments are portable and therefore potentially field-deployable. Especially the small measurement cavity size (35 mL) of wavelength-scanned cavity ring-down spectroscopy (WS-CRDS) instruments makes them ideal for lab-scale experimental setups as well as for small sensor designs. The spread of laser-based instruments has therefore stimulated recent developments of a number of in situ methods for direct measurements of water stable isotopes in various fields. Precipitation measurements were carried out via gas-permeable ePTFE surgical tubings (Munksgaard et al., 2011). Soil column breakthrough curves (Herbstritt et al., 2012) as well as analyses of precipitation and canopy throughfall in parallel were achieved via small hydrophobic membrane contactors (Herbstritt et al., 2019). The isotopic composition of pore water was analysed in lab-scale experiments via hydrophobic microporous tubings (Rothfuss et al., 2013) as well as in natural soil profiles with custom-made hydrophobic porous in situ water isotope probes (WIPs) (Volkmann and Weiler, 2014). Similar in situ probes were also used in tree stems in labelling experiments to analyze the isotopic composition of xylem sap (Volkmann et al., 2016a; Seeger and Weiler, 2021). The 'stem borehole method' (Marshall et al., 2020; Kühnhammer et al., 2022) is an alternative way to obtain in situ samples of tree xylem water vapor. Measurements of the isotopic composition of transpired water were conducted using leaf chambers (Wang et al., 2012; Dubbert et al., 2014) or whole-plant chambers (Volkmann et al., 2016b).

The isotopic composition of the liquid water of interest in all these in situ studies was inferred from sampling and measuring a corresponding vapor phase. Water vapor of interest was either withdrawn directly e.g. from soil profiles, out of tree boreholes, or exchange and equilibration with a carrier gas through different types of hydrophobic membranes were facilitated.

However, operating laser-based analyzers at the study site requires power-consuming, heavy and expensive equipment to be brought to the field with the risk of damages and the

disadvantage of relative immobility. Consequently, rough or remote terrains as well as spacious experimental designs, exceeding possible tubing lengths between in situ measurements and the isotope analyzer are virtually excluded with this approach. First attempts to overcome these obstacles by collecting discrete vapor samples under such circumstances into glass bottles were recently presented. In both approaches, the sampled vapor had to be diluted continuously during measurement, to compensate for negative pressure when sucked into the analyzer by releasing e.g. dry air into the rigid, fragile sampling flasks. The first approach (Havranek et al., 2020) can be seen as a proof of concept. In a follow-up, Havranek et al. (2022) describe a field application of their setup which requires considerable financial resources for the components in use, extensive technical know-how for construction as well as substantial effort for field installation of a limited number of flasks. Their recommended operation procedure requires long flushing times, leading to filling times per flask of more than one hour, which strongly reduces the sample throughput and thus the achievable temporal resolution. The second approach (Magh et al., 2022) partly resolved these issues but still relies on specialty tools, lacks reproducibility due to the small volume of sampled vapor and currently does not provide the data accuracy needed for natural abundance isotope assays. Further, a sophisticated calculation procedure is necessary for both approaches (Havranek et al., 2020; Magh et al., 2022) to remove the effects of the initial pulse of water vapor during the start of the measurement phase. This initial pulse is mixed with pre-sample vapor which clearly biases the obtained isotope data.

Therefore, the aim of this study is to develop a technique to collect discrete vapor samples in the field for subsequent lab-based analyses, that overcomes the aforementioned problems while still ensuring handiness as well as cost and time efficiency. Specifically, we identified a way to prepare and measure calibration and validation vapor standards, necessary for processing unknown, field-collected vapor samples. For this purpose, we varied the applied

gas flowrate through established non-destructive in situ water isotope probes (WIPs) to improve per-sample time consumption. We tested the diffusion-tightness and inertness of various commercially available gas sampling bags as well as custom-made inflatable containers comprising different materials and closing mechanisms. We identified the best performing bags and coupled it with WIPs in order to collect and temporarily store discrete vapor samples prior to lab-based isotope analyses. Also, we identified necessary preparatory steps to optimize the reproduction of in situ data.

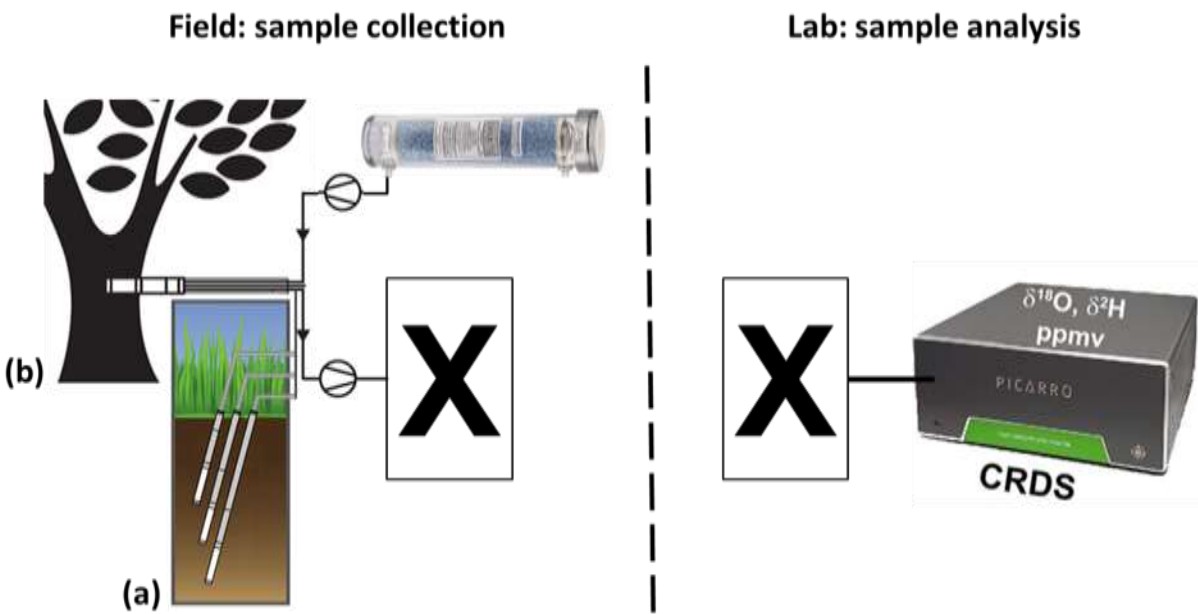

Figure 1. Schematic of projected vapor sampling via in situ (a) soil- and (b) xylem-water isotope probes (modified from Volkmann & Weiler, 2014 and Volkmann et al., 2016a). The left part describes the intended field setting while the right part describes the intended laboratory setting: sampled vapor is first filled into appropriate, to-be-identified containers (X) and later analyzed in the lab via CRDS.

**2 Methodology**

**2.1 Effect of changing gas throughflow rates on the isotopic composition**

In the first part of this study we tried to optimize sample filling times. For this purpose we investigated the effect of varying gas flowrates through the to-be-employed in situ water

isotope probes (WIPs). Originally, the size of the probes was optimized i.e. the contact area of the membranous tip, to facilitate isotopic equilibrium when the applied flow rates had been set to match the analyzer's demand. This prerequisite becomes obsolete for the collection of discrete vapor samples. In the case of discrete vapor sampling through WIPs into air-tight containers, different, arbitrary flowrates can be selected, which inversely affect the filling times of the containers. It is important that the contact area relevant for vapor collection is consistent for all sample collections as isotope equilibrium cannot be expected anymore when the applied gas flowrates exceed the ones originally recommended (Volkmann & Weiler, 2014). For repeated analysis of a single vapor sample we found a total gas sample volume of 0.5 - 1 L to be sufficient (Gralher et al., 2021). Further, we aimed at sampling times in the field of no more than 5 minutes per sample. Therefore, flow-rates of both dry air through the WIP and the corresponding gas sampling rate into the containers were increased stepwise in a lab experiment up to 150 mL/min which would yield 0.75 L sample volume after a filling time of 5 min. We also tested the effects of omitting the originally proposed dilution (Volkmann & Weiler, 2014) when increasing the throughflow rates. In order to keep the gas flow inside the WIP balanced between the inflowing carrier gas and the outflowing sampling gas and in doing so avoid over- or under-pressure, the sampling rates had to match the throughflow rates. For these tests a WIP was installed in an evaporation-shielded box filled with moist sand with a water stable isotopic composition of -9.64‰ and -66.84‰ for $\delta^{18}O$ and $\delta^2H$, respectively, referenced to the VSMOW-SLAP scale (Craig, 1961) and kept at a constant temperature of 20.8 °C. For precise isotope measurements we used dry synthetic air ('zero air') as carrier gas.

Precise flowrates of the synthetic air from the pressurized gas bottle into the WIP were facilitated by a digital mass flow controller (PN 35828, Analyt MTC, Müllheim, Germany), while sampling at the same flowrate was facilitated by a small air pump (PN LP27-12, Pollin

Electronic GmbH, Pförring, Germany) where the pumping rate can be controlled manually via the applied voltage and controlled with a mass flow meter (PN 35808, Analyt MTC). The fractions of sampling rates exceeding the analyzer demand of ~ 35 mL/min were vented to air through an open split near the sample inlet port of the isotope analyzer (L2120-*i* or L2130-*i*, Picarro Inc., Santa Clara, CA, USA). The analyzer provided quasi-continuous (0.5 Hz) readings of water vapor mixing ratio (in ppmv), oxygen and hydrogen isotope readings (in ‰), and the spectral parameters 'h2o_vy' indicating mixing with ambient air and 'organic_MeOHampl' indicating spectral interference from the bag material, which we also collected from room air on every day of bag measurements. This setup allowed for facilitating the demanded low and constant stream of gas to the analyzer while at the same time arbitrarily varying the gas flow through the attached WIP.

## 2.2 Material selection

### 2.2.1 Diffusivity and spectral interference

Rigid glass or steel bottles and cylinders conventionally used for gas sampling were excluded in our approach, since constant flow through the WIPs was needed in our setup and under-pressure at the analyzer during measurements has to be avoided. Commercially available gasbags with large volumes (10 L) are available with reusable gas valves in opposite to bags with smaller volumes (0.5 L – 2.5 L), which come with septa or other degrading closures. Hence, none of these combinations were suitable for our purposes, either due to their size or due to the type of closure. Other readily available gas sampling bags made of PTFE or laminated aluminum foil from a different supplier were available in appropriate sizes but were available only in minimum order quantities of 10. Their per-order costs ranged between €200 and €350 depending on bag material which were prohibitive for our purposes. Therefore, we investigated three different inflatable bags of different materials and reasonable sizes and combined them with different reclosable caps and valves (Table 1). Additionally, a 2.5 L

commercially available gasbag was equipped with a stainless steel screw-lock valve and also tested. Specifically, we evaluated whether they were sufficiently reliable in terms of diffusion-tightness and chemical inertness, also focusing on easy handling. The criteria for identifying reliable bags or bag material are summarized in a best-practice protocol (SOP) in the Appendix of this manuscript.

The different bag types were filled with pure $N_2$ (purity 99.996%) and analyzed immediately thereafter with a CRDS isotope analyzer (L2130-*i*, Picarro Inc., Santa Clara, CA, USA). We tested PE spoutbags (code 'PE-Sp', PN 1055) and single layer metalized spoutbags (code 'Al-Sp', PN 1050, both available from Daklapack Europe, Oberhausen, Germany), of which we replaced the original PE spout caps by caps with rubber septa. Filling and continuous vapor isotope analyses were facilitated through these septa via an infusion needle (ID = 1 mm) assembled to a 1/8" perfluoroalkoxy alkane (PFA) tube. We also tested two kinds of 3-layer metalized zip bags with fill volumes of 1 L (PN: CB400-420BRZ, color: red) and 0.5 L (PN: CB400-310GZ, color: gold, both available from Weber Packaging GmbH, Güglingen, Germany). They were heat-sealed and equipped with silicone blots on the outside, which served as septa after 2 days of drying (code 'Al3-Sil' for the 1 L bag, code 'Al3g-Sil' for the 0.5 L bag). The bags were filled and their content withdrawn for isotope analysis through the silicone septa again via an infusion needle assembled to a 1/8" PFA tube.

To improve handiness and simplify filling and sample analysis when using the 3-layer metalized bags, we tested two different types of valves as alternatives to our custom-made silicone septa. With a punching tool, a hole was applied to each bag for the respective diameter of the screw connections. We fixed small pneumatic brass couplings (PN KDG M5 NW2,7, Landefeld, Kassel, Germany) on the 3-layer metalized 1 L zip bags (code 'Al3-PC'), which were then heat-sealed. The respective plug connector (PN KSGI M5 NW2,7, Landefeld, Kassel, Germany) was connected to a 1/8" PFA tube for filling and analysis. Also

stainless steel screw-lock gasbag valves with 6 mm hose fittings (PN 11701150, Linde, Pullach, Germany) were mounted on 0.5 L and on 1 L 3-layer metalized zip bags (codes 'Al3s-GbV' and 'Al3-GbV') after punching a 10 mm hole. For increased leak-tightness we mounted an additional, custom-made rubber washer between the valve and the inner wall of the bags which were then heat-sealed. For filling as well as sample analysis the 6 mm hose fitting was adapted to a 1/8" PFA tube.

Table 1. Bag- and seal-type combinations and their properties tested for vapor sampling.

| Code | Material | Bag style | Vol | Closure, cap or seal type | Inflation, filling, sampling via… |
|------|----------|-----------|-----|----------------------------|-----------------------------------|
| PE-Sp | PE | spoutbag | 1 L | PE screw cap w/ rubber septum | Infusion needle and 1/8" PFA tube |
| Al-Sp | Al | spoutbag | 1 L | PE screw cap w/ rubber septum | Infusion needle and 1/8" PFA tube |
| Al3g-Sil | Al 3ply gold | Zip bag | 0.5 L | Silicone blot as septum | Infusion needle and 1/8" PFA tube |
| Al3-Sil | Al 3ply red | Zip bag | 1 L | Silicone blot as septum | Infusion needle and 1/8" PFA tube |
| plastigas® | Al 3ply silver | | 2.5 L | Stainless steel screw-lock valve | Hose fitting adapted to 1/8" PFA tube |
| Al3-PC | Al 3ply red | Zip bag | 1 L | Pneumatic brass coupling | Plug connector and 1/8" PFA tube |
| Al3-GbV | Al 3ply red | Zip bag | 1 L | Stainless steel screw-lock valve | Hose fitting adapted to 1/8" PFA tube |
| Al3s-GbV | Al 3ply silver | Zip bag | 0.5 L | Stainless steel screw-lock valve | Hose fitting adapted to 1/8" PFA tube |

To detect gradual diffusive exchange of the bag content with ambient air or outgassing from the employed material, 3 to 5 replicates of each bag/valve combination were flushed with $N_2$ and evacuated twice before they were again filled with pure $N_2$. They were then stored at ambient temperature and repeatedly analyzed over the course of four weeks. Measurement frequency was every two to three days during the first two weeks and one final time at the end of the fourth week, unless a bag/valve combination was found unsatisfactory earlier.

### 2.2.2 "Climate chamber" experiment

To further test the vulnerability of projected discrete vapor samples, i.e. to test if relative humidity outside of the bags can exchange with and thus flaw the sample inside the bag, we designed a small "climate chamber" which consisted of a plastic box (inner dimensions 57 cm × 37 cm × 32 cm) covered by a plastic lid with all holes and slits taped. We prepared six 'Al3s-GbV' bags filled with vapor from the same source (described in detail in 2.3). Three

bags were placed in the box straightforward while the other three were inserted in metal cans of the bags' size prior to placement in the box. Such metal cans are normally used for the

 transport and storage of glass bottles containing liquid water sampled for dissolved gases analysis. They are considered diffusion-tight when closed and sealed by means of metal lids, rubber seal rings, and metal clasps as we did. Inside the box we also placed an open bowl of water (ca. 350 mL) to quickly reach and then maintain a relative humidity near 100% over the course of the test which lasted three weeks. The box was deposited in the basement of our

 laboratory building to facilitate fairly stable temperature conditions. Temperature (°C) and relative humidity (%) inside the box were recorded every ten minutes with a CS215 probe connected to a CR200 logger (both from Campbell Scientific, Logan, UT, USA). These data were then converted to water vapor mixing ratios using Magnus' equation (Foken, 2008). Vapor concentration (ppmv) and isotope ($\delta^{18}O$, $\delta^2H$) data from the vapor source used for

 filling the bags, from the inside of the box after 20 days, and from all canned and un-canned bags after 20 days of storage inside the box were collected with a Picarro 2120-$i$.

**2.3 Field trial**

We tested the reliability of the projected sampling procedure first in a laboratory experiment and later in the field using 'Al3-GbV' bags. For this purpose, four evaporation-shielded boxes

 (V = 18 L) with moist sands with different water isotopic compositions were prepared and a WIP (Volkmann & Weiler, 2014) was installed in each of the boxes to sample their soil water vapor. We used a low-weight sampling setup that provided a constant air flow of 150 mL/min with the small pumps described above (section 2.1). The incoming stream of ambient air was dried by a 'Drierite' drying column (PN 26800, W. A. Hammond DRIERITE Co. Ltd., Xenia,

 OH 45385, USA) and directed through the throughflow line into the porous tip of the WIP. The vapor generated inside the WIP was withdrawn through the sampling line by a second small pump with the same gas flow-rate. The stream of sampled vapor was directed to the

isotope analyzer (demanding 35 mL/min), using an open split for excess vapor (115 mL/min)
near the analyzer's sample inlet port. Immediately after the direct measurement, the analyzer

was disconnected and the setup was used for directing the entire gas stream into the bags.
Additionally, ambient vapor data were collected with the isotope analyzer. Replicates of
sampled vapor were filled in bags, which were then analyzed two hours later. Calibration was
facilitated using in situ and bag measurements of those samples displaying the highest and the
lowest $\delta^2H$ values, treating these in situ values de facto as standards. This selection was

maintained for calibration of $\delta^{18}O$ values, too, although that meant that the so-selected
standards did not "bracket" the to-be-calibrated "samples" as is common best practice. The
precision would be the standard deviation of repeated calibrated isotope readings of the
samples displaying intermediate $\delta^2H$ readings. The accuracy would be the deviation of the
calibrated mean of replicates of the respective in situ measurements. Note that this led to the

reproduction (validation) of the intermediate in situ values rather than liquid water values
referenced to the VSMOW-SLAP scale (Craig, 1961). The derivation of liquid water values
from vapor isotope observations has been described in Volkmann & Weiler (2014).

**2.4 Reusability of sampling bags**

**2.4.1 Flushing attempts**

To test the reusability of the bags, we applied the following flushing procedure. 'Al3-GbV'
and 'Al3s-GbV' bags were filled with pure $N_2$, evacuated immediately thereafter with a
LABOPORT® diaphragm vacuum pump (N810.3 FT.18, KNF Neuberger GmbH,
Munzingen, Germany) and filled again with pure $N_2$. On the next day they were evacuated
again, filled with pure $N_2$ again and evacuated one final time. After these preparatory steps, 3-

5 bag replicates were used for sampling vapor from sources both isotopically different and
identical to the ones that had been sampled before with the respective bags. The setup used
for this purpose was identical to the one described in Section 2.3 except that different

diaphragm gas pumps (BOXER 22K, Boxer, Ottobeuren, Germany) were used. Vapor concentration and isotope data of the bags were recorded with a Picarro L2120-$i$ 1, 3, and 7

290    days after vapor sampling.

### 2.4.2 Conditioning

We additionally tested the reusability of the bags by comparing two ways of conditioning previously used sampling bags. The first way was by filling a batch of bags with dry synthetic air, leave them filled for at least one day, analyze their vapor concentration and isotope

signature with a Picarro L2120-$i$, evacuate them, fill them again, and repeat this cycle several times. The second way was identical except that moist, isotopically homogeneous air was used for filling and priming. In both cases the absolute vapor concentrations and the standard deviations (SD) of isotope readings from repeated batch measurements were considered as predictor for conditioning efficiency. Efficiency was then scrutinized by using the so-

conditioned bags for collecting vapor samples from isotopically diverse sources (setup details in 2.3) followed by repeated analyses over the course of up to seven days. Mean vapor isotopic compositions of these sources were -20.41, -29.32, and -37.24 for $\delta^{18}O$ and -84.95, -139.53, and -195.77 for $\delta^2H$. Again, calibration was facilitated using bag measurements of those samples with the highest and lowest isotope values and their respective in situ

measurements. We report precision and accuracy as quality measures of the calibration process. The reported precision is the SD of the repeated calibrated isotope readings of the intermediate source which we consider as validation standard while the reported accuracy is the deviation of the calibrated mean of repeated measurements from the respective target value which are the isotope readings from the respective in situ measurements.

**3 Results**

**3.1 Effect of changing gas throughflow rates on the isotopic composition**

With increasing flow-rates through the WIP, the vapor content originating from the source water decreased in the sampling gas (Fig. 2). At the same time, the isotopic composition for both isotope ratios under investigation changed, indicating that no equilibrium between the source water and the provided dry gas stream was established inside the WIP. Such kinetic fractionation effects could be observed at all flow-rates exceeding the originally proposed equilibrium flowrate from Volkmann & Weiler (2014). They were stronger for $\delta^{18}O$ than for $\delta^2H$. The induced decreasing vapor contents were strongly correlated with decreasing isotopic signatures. In the case of the WIP operated with proportional dilution and throughflow rates, vapor content ranged from 10325 to 14432 ppmv, while the concurrent uncalibrated isotope readings were in a range of -18.92‰ to -30.99‰ in the case of $\delta^{18}O$ and -136.85‰ to -143.41‰ in the case of $\delta^2H$. The heaviest values of these ranges correspond to the settings originally proposed by Volkmann & Weiler (2014). These changes correspond to change rates of 0.241‰/(mL/min) for $\delta^{18}O$ and 0.131‰/(mL/min) for $\delta^2H$. Similar observations were made for the case when dilution was set to zero at higher flowrates. Here, the vapor content ranged from 12256 to 20041 ppmv while the uncalibrated isotope readings varied between -30.35‰ and -39.13‰ in the case of $\delta^{18}O$ and between -143.23‰ and -146.95‰ in the case of $\delta^2H$. Here, the changes correspond to change rates of 0.117‰/(mL/min) for $\delta^{18}O$ and 0.049‰/(mL/min) for $\delta^2H$. Vapor was always sampled from the same liquid water source which had an isotopic composition of -9.64‰ and -66.84‰ for $\delta^{18}O$ and $\delta^2H$, respectively, referenced to the VSMOW-SLAP scale (Craig, 1961). All coefficients of determination (R²) between vapor content and isotope readings were greater than 0.99. At the target flow-rate of 150 mL/min throughflow and 0 mL/min dilution vapor content was close to 12,000 ppmv at 20.8 °C (Fig. 2).

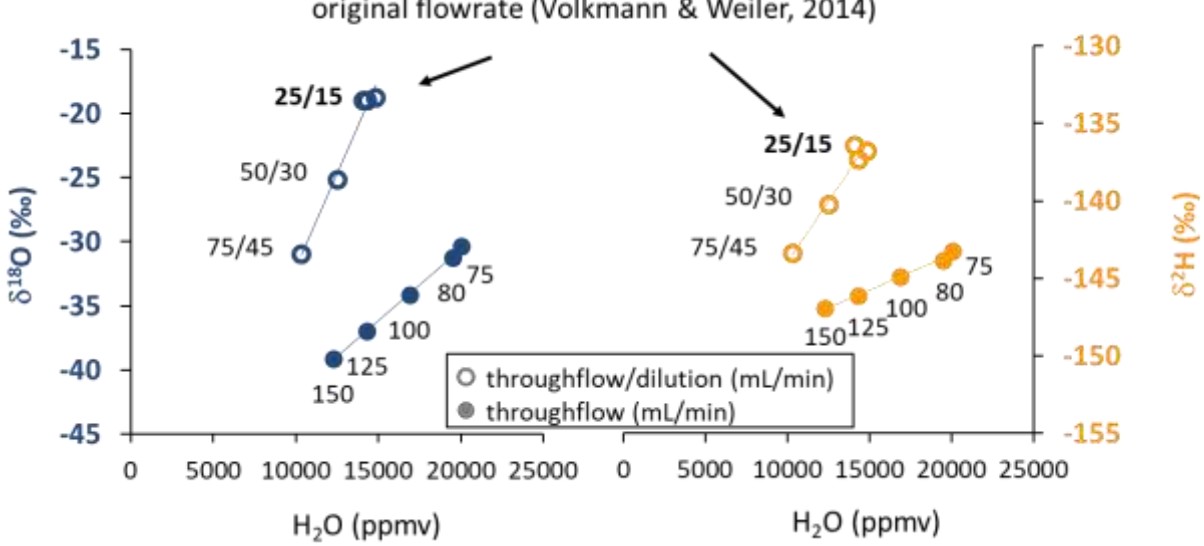

Figure 2. Scatterplot of water vapor isotopic composition (left: $\delta^{18}O$, right: $\delta^2H$) and vapor content obtained by varying gas throughflow rates with (open symbols) and without (closed symbols) dilution at a constant temperature, illustrating the kinetic fraction effects with increasing flowrates through the probe. The numbers refer to the respective gas flowrates applied as throughflow/dilution or throughflow-only during the tests. The flowrates originally proposed by Volkmann & Weiler (2014) facilitated equilibrium according to the design of the probe's tip size and the known instrument flowrate.

### 3.2 Material selection

#### 3.2.1 Diffusivity and spectral interference

**Diffusion-tightness and long-term storage effects.** During the test for diffusion-tightness the water vapor content readings of the ambient air in the lab were in a range of 9000 – 18000 ppmv while inside the bags they were initially close to zero due to pure $N_2$ inflation. Over time, these vapor pressure gradients were gradually levelled out with different rates. This is qualitatively evident from the different slopes of the dashed lines (Fig. 3). PE spout-bags 'PE-Sp' displayed the highest increase when after three days the vapor content had already increased by ~7000 ppmv. Vapor content readings in the metalized spout-bags 'Al-Sp' were ~4000 ppmv after three days. 3-layer metalized bags displayed the lowest vapor increase rate. Design and thus air-tightness of these bags differed only with the type of septum or valves

used. The vapor content readings in the bag-types with the silicon blot ('Al3-Sil') were

~1200 ppmv after four days.

Only for the better performing bag/valve combinations ('Al3-PC', 'Al3-GbV' and 'Al3s-GbV') we extended the vapor content measurements up to four weeks. After this period, mean vapor content readings in 'Al3-PC' and 'Al3-GbV' bags were ~3000 ppmv and ~1450 ppmv, respectively. Mean vapor content readings in 'Al3s-GbV' bags were ~420 ppmv after

two weeks, which was the final value possible for this bag type due to its smaller volume.

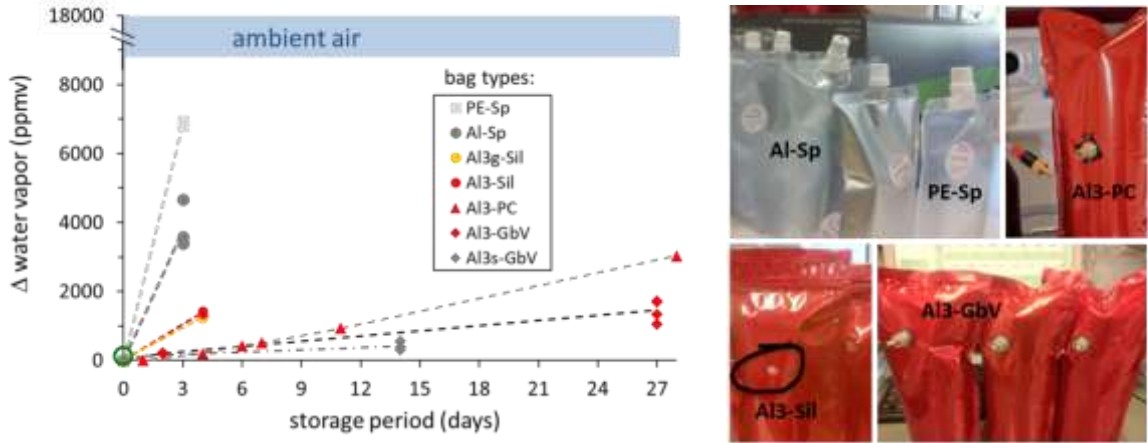

Figure 3. Left panel: Time series of vapor content readings inside different bag types (closed symbols), initially filled with pure $N_2$ (green open circle). Right panel: Example pictures of tested sampling bags.

**Spectral interference of outgassing material.** In two bag types, deviations of the spectral parameters from the pure $N_2$ signal were found, which were also correlated with the respective isotope readings. The spectral line width variable indicative for gas composition ('h2o_vy') is $0.4309 \pm 0.0015$ ppm on our L2130-$i$ analyzer for air containing oxygen at atmospheric levels, while it is $0.4563 \pm 0.0049$ ppm for pure $N_2$. Vapor concentration of bag-

type 'Al3g-Sil' containing pure $N_2$, was below 2000 ppmv and should therefore plot at a h2o_vy value of about 0.46 ppm like the bags 'Al3-Sil' (same material but different color), but evolved towards 0.43 ppm. Simultaneously, apparent enrichment in heavy isotopes was

observed in 'Al3g-Sil' bags with an increase of around 80‰ in $\delta^{18}O$ and 150‰ in $\delta^2H$, compared to samples stored in 'Al3-Sil' bags.

In the commercially available 2.5 L plastigas® bag equipped with a gasbag valve (PN 11701150, Linde) (combination not featured in Fig. 3), we observed an apparent depletion in $\delta^2H$ from -190‰ to -305‰ after 24h and to -360‰ after 72h. At the same time a spectral variable recorded on the L2120-*i* analyzer, indicating potential contamination with organic compounds ('organic_MeOHampl'), increased to 0.00760 ± 0.00014 after 24h and to 0.01005

± 0.00012 after 72h. The initial value was 0.00095 ± 0.00026 which was also observed in ambient air. We therefore also excluded 'Al3g-Sil' and 'plastigas®' bags from further testing due to the observed spectral interferences. Further tests and isotope samplings in our study were conducted with 'Al3-GbV' or 'Al3-GbV' bags only. The performance of the different bag materials in the tests is summarized in Table 2. The protocol for testing bags and material

properties and the respective target values for passing the tests can be found in Appendix A.

Table 2: Performance of tested bag types. Test passed is indicated by ✔, failed by **X**, not conducted by –.

| Code | Diffusivity | Spectral Interference | Isotopic Validation |
|---|---|---|---|
| PE-Sp | **X** | – | – |
| Al-Sp | **X** | – | – |
| Al3-Sil | **X** | – | – |
| Al3g-Sil | **X** | **X** | – |
| plastigas® | **X** | **X** | – |
| Al3-PC | (✔) * | ✔ | – |
| Al3-GbV | ✔ | ✔ | ✔ |
| Al3s-GbV | ✔ | ✔ | ✔ |

* coupling corrosive

### 3.2.2 "Climate chamber" experiment

Temperature was quite stable inside the climate chamber, as intended. It ranged between 18.1°C and 16.3°C. Relative humidity rose to > 97% within 6 hours and maintained on average at 99.5% throughout the remaining observation period, which translated to vapor mixing ratios of 19972 ± 519 ppmv. Mean vapor contents from the source were 14365 ppmv during filling and 12574 ppmv inside all bags after 20 days. Differences in mean vapor contents between canned and un-canned bags were smaller than the respective variations within the two batches. Isotope readings of all bags appeared to be enriched by 2.9‰ in $\delta^{18}$O and 15.0‰ in $\delta^2$H relative to the source, again with negligible differences between the two batches (data shown in Supplement Fig. S1).

### 3.3 Field trial

When comparing all isotope data of the direct in situ sampling from the sand boxes in the field with the data from vapor sampled into the bags and measured 2 hours later, we found no systematic bias towards congruent enrichment or depletion in heavy isotopes. Instead, all raw isotope data of the bag measurements appeared to be shifted towards ambient air values by -0.6 to +1.6‰ for $\delta^{18}$O and +4 to -5‰ for $\delta^2$H relative to their respective in situ measurements. We used isotope readings of in situ and bag measurements from the samples displaying extreme $\delta^2$H values for calibration. Thereby, in situ values of the intermediate samples could be reproduced by calibrated bag measurements with a precision of 0.15‰ and an accuracy of 0.8‰ for $\delta^{18}$O. The respective numbers for $\delta^2$H were 0.76‰ and 2.89‰ (Fig. 4).

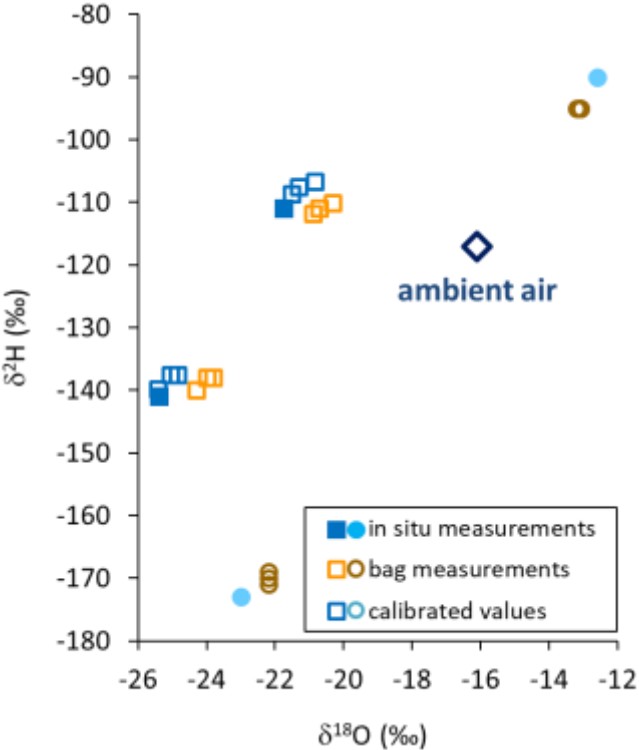

Figure 4. Isotope data from in situ measurements (filled symbols), raw bag measurements (orange and brown open symbols), and calibrated in situ data calculated from bag measurements (blue and cyan open symbols) using in situ (cyan closed symbols) and raw values (brown open symbols) of sources displaying extreme $\delta^2H$ values. Ambient vapor at that day is displayed as a blue open diamond.

### 3.4 Reusability of sampling bags

### 3.4.1 Flushing attempts
Generally, isotope data obtained from repeated measurements of the bags unanimously trended towards the values recorded previously from the respective sample bags. These trends increased over time and changes were proportional to the differences between current in situ and previous bag measurements on the individual bag level. Such memory effects persisted although the bags had been evacuated and flushed three times with dry $N_2$ prior to vapor sampling. The trends were consistent for all tested bag types and appeared to be independent from concurrent ambient air values for both isotope ratios investigated (data shown in Appendix Fig. A1).

### 3.4.2 Conditioning

The two different conditioning methods (dry and moist) applied to previously used sample bags yielded contrary results. Conditioning with dry synthetic air caused vapor content readings to decrease stepwise down to 324 ppmv (Figure 5a) while isotope signatures became more enriched (isotope data shown in Supplement Fig. S2). Their SDs generally decreased but remained above 2.9‰ for $\delta^{18}O$ and 18.8‰ for $\delta^2H$ (Figure 5b). In contrast, conditioning with moist air resulted in vapor content readings to decrease to 6740 ppmv, which was in the order of the level of conditioning. Isotope signatures of so-primed bags clustered around the conditioning values (data shown in Supplement Fig. S3) with SDs decreasing to 0.05‰ for $\delta^{18}O$ and 1.07‰ for $\delta^2H$ after 4 to 5 steps (Fig. 5b).

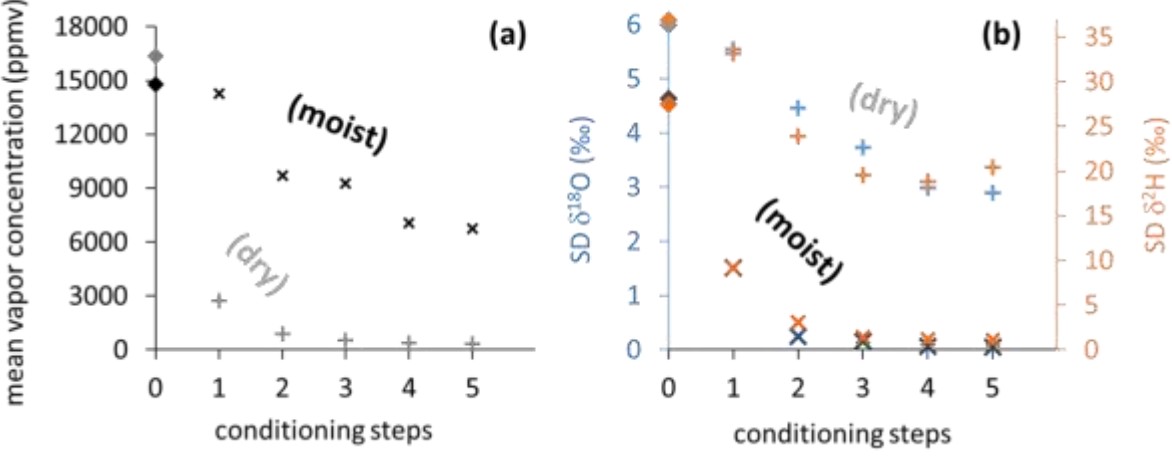

Figure 5. Vapor content readings (a) and SDs of isotope readings (b) from vapor sampling bags, stepwise conditioned with dry synthetic air (light plus symbols) and moist air (dark cross symbols). Light and dark diamond symbols on vertical axes represent mean values of pre-conditioning sample measurements.

Re-using the conditioned bags for sampling vapor from isotopically diverse sources also yielded contrasting results. SDs of isotope readings in bag replicates after dry conditioning were larger than after moist conditioning. Repeated measurements generally resulted in a decline of measurement precision and accuracy of mean isotope values (Fig. 6). Comparing in

situ values with bag measurements yielded no consistent picture in the case of dry conditioning (Fig. 6a) whereas in the case of moist conditioning a bias towards the conditioning values became evident which increased over time (Fig. 6b). One day after filling, raw isotope data of the bag measurements deviated by -0.6 to +1.6‰ for $\delta^{18}O$ and +4 to -5‰ for $\delta^2H$ relative to their respective in situ measurements. Calibration for reproduction of the intermediate in situ values worked better for moist- than for dry-conditioned bags. This refers to the precision as well as to the accuracy (Table 3).

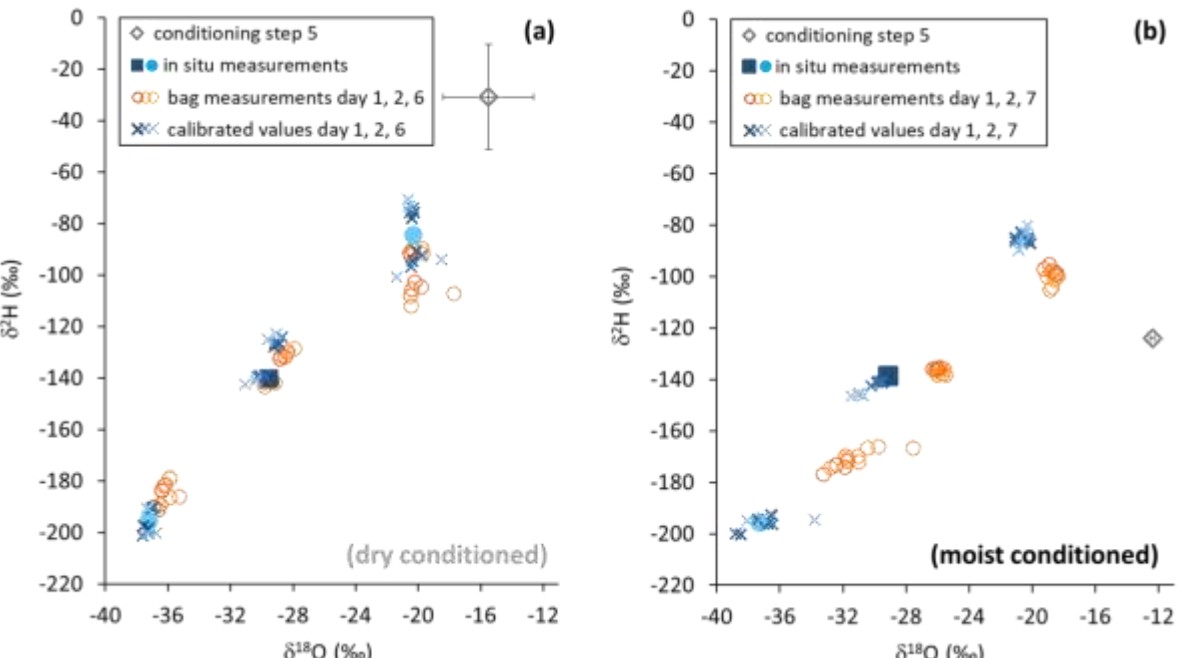

Figure 6. Isotope data from in situ measurements (filled symbols), gasbag measurements (open circles), and in situ data calculated from gasbag measurements (asterisks) after conditioning with a dry (a) and moist atmosphere (b). Pre-sampling levels are represented by gray diamonds. Note that error bars (black) are smaller than the symbol in the case of moist conditioning (b).

Table 3. Conditioning effects on vapor stable isotope measurements from re-used sampling bags.

| Day after sampling | Analysis iteration | Dry atmosphere conditioning δ$^{18}$O (‰) | | δ$^2$H (‰) | | Moist atmosphere conditioning δ$^{18}$O (‰) | | δ$^2$H (‰) | |
|---|---|---|---|---|---|---|---|---|---|
| | | precision | accuracy | precision | accuracy | precision | accuracy | precision | accuracy |
| 1 | 1 | 0.58 | -0.10 | 6.71 | -7.16 | 0.25 | 0.41 | 0.41 | 1.93 |
| 2 | 2 | 0.70 | -0.02 | 8.14 | -7.89 | 0.30 | 0.70 | 0.75 | 2.59 |
| 6 | 3 | 0.87 | 0.51 | 10.36 | -7.40 | | | | |
| 7 | 3 | | | | | 0.30 | 1.91 | 0.40 | 7.18 |

## 4 Discussion

**Gas flowrate effects.** The goal of this study was to facilitate storage of in situ sampled vapor without sensitive and costly analytical equipment in the field. The main applications we had in mind were high frequent, minimally invasive in situ isotope measurements of water vapor in soils and plant tissue as recently performed by e.g., Volkmann & Weiler (2014), Volkmann et al. (2016a), Seeger & Weiler (2021), or Gessler et al. (2021). Their setups were based on applying stable, yet low gas flowrates to their WIPs. These flowrates resulted in continuous vapor samples assumed to be in isotopic equilibrium with the liquid water of interest. They had been carefully adjusted to the employed analyzers' demands. However, in the intended absence of an analyzer this necessity becomes obsolete. Therefore, we first tested the effect of higher gas flowrates on the obtained vapor isotope signatures. We did so in order to facilitate shorter filling times that would allow for higher filling frequencies of any containers of sufficient volume (0.5 – 1 L) needed to collect and store vapor samples prior to lab-based analyses.

Incidentally, we also tested the necessity of isotope equilibrium for water vapor sampling using WIPs. We found that also under non-equilibrium conditions, in-situ isotope values could be reproduced with a very good precision and accuracy. We attribute this to the fact that all employed WIPs had been constructed equal in size. This resulted in consistent gas transit and vapor pick-up times inside the tips as well as consistent relevant soil contact area which is the vapor collection area. This expands the findings of previous studies which aimed at facilitating equilibrium conditions when sampling water vapor from soils or stem boreholes

(Volkmann & Weiler, 2014; Marshall et al., 2020). This also means that our approach may not be suitable for application together with the stem borehole method as the tree stem diameters will likely be variable thus not allowing for consistent gas transit and vapor pick-up times. By using identical WIPs throughout our experiment as well as for standards and samples in the foreseen application, potential isotope fractionation effects induced by the membrane are eliminated in the calibration process. Such membrane-induced fractionation effects have been observed before during across-membrane collection of vapor for liquid water isotope determination (Herbstritt et al., 2012). Further, non-equilibrium conditions as encountered in this study did not require any extra correctional efforts. Mathematically, the applied calibration routine is identical to the case of the direct vapor equilibration method (Wassenaar et al., 2008; Hendry et al., 2015, Gralher et al., 2021) or even routine, automated liquid water isotope analyses (Werner & Brand, 2001).

Although pure $N_2$ as a carrier gas would have been cheaper, we used synthetic air in order to maintain a consistent nitrogen-to-oxygen mixing ratio. This helped to avoid gas matrix effects previously demonstrated for CRDS instruments (Gralher et al., 2016). We found that gas flowrates higher than the ones previously applied (Volkmann & Weiler, 2014; Volkmann et al., 2016) immediately resulted in incomplete equilibrium between the liquid water under investigation and the obtained vapor sample (Fig. 2). This became evident by decreasing, below-saturation vapor contents as well as lighter isotope readings. On the other side, we found that even the highest gas flowrates applied in this study (150 mL/min) still yielded vapor concentrations (~12000 ppmv @ 21°C) that are high enough to be within the analyzer's optimum measurement range and thus enable sufficiently precise isotope measurements for resolving natural variations (URL1). At the same time, a dilution of the obtained vapor stream is obsolete under such settings as the low obtained vapor concentrations impose no risk of condensation. Of course, field sites being sampled for vapor while enduring temperatures that

are much higher than the lab temperature might yield too high vapor concentrations. Then even higher flow rates or the re-application of a dilution flow is necessary for compensation to avoid condensation and thus un-correctable isotope fractionation. This implies that like for other indirect, minimally invasive methods (Volkmann & Weiler, 2014; Magh et al., 2022)

knowledge about the temperature or maintenance of its consistency at the points of vapor sampling is mandatory for interpreting the obtained isotope data. Cases where temperature differs considerably among sampling sites and/or the site of calibration standard preparation require additional correction schemes considering the temperature-dependencies of water-vapor isotope fractionation. This could be facilitated either via mathematical approaches

based on the dependencies described e.g., by Majoube (1971) or via empirical approaches derived from sets of calibration standards that were collected while intentionally being subjected to different, controlled temperatures. Calibration standards can be prepared e.g., by installing WIPs in evaporation-shielded, sand-filled boxes wetted with water of known isotopic composition as detailed in Volkmann & Weiler (2014).

The changes in isotope readings and thus deviations from equilibrium were smaller for $\delta^2$H than for $\delta^{18}$O in absolute numbers (10.1‰ vs. 20.21‰, respectively) and even more so in relation to naturally occurring isotope variations, where changes usually exhibit an 8:1 ratio in meteoric waters (= slope of the GMWL, Craig, 1961). Therefore, in situ isotope assays relying on discrete vapor sampling for later analysis with a setup similar to the one foreseen

here, have to ensure precise control of the applied gas flowrates for samples and co-measured standards in order to comply with the paramount Principle of Identical Treatment (Werner and Brand, 2001). If the isotope ratio under investigation is optional, we recommend interpreting hydrogen rather than oxygen isotope ratios given the lower susceptibility regarding gas flow rate effects and thus a more favorable signal-to-noise ratio. In this context,

it is quite convenient that labeling studies are more cost-efficient when using deuterium as tracer rather than oxygen-18 (Magh et al., 2022).

Flowrates through the probes, exceeding previously recommended settings (Volkmann & Weiler, 2014) immediately resulted in changes of vapor concentration and isotope readings (Fig. 2). We therefore argue that neither the previous recommended nor our newly selected settings result in complete isotopic equilibrium. Hence, both settings require precise control of flowrates as any uncertainty in flowrate settings translates to systematic errors in isotope readings. These errors are considerably higher at lower flowrates and higher for $\delta^{18}O$ than for $\delta^2H$. Aiming at lower flowrates in order to reliably achieve equilibrium readings appears impractical for two reasons. Firstly, the employed analyzer's gas flow demand defines the minimum total gas flow. A workaround applying higher dilution flowrates would yield varying vapor concentrations and thus vapor concentration effects as can be seen from the different isotope readings observed for 75mL throughflow rate with or without dilution (Fig. 2) thus potentially introducing additional errors. And secondly, applying lower flow rates contradicts the study aim of shorter filling times and thus higher sample collection frequencies and achievable temporal resolution.

**Material and closure type selection.** Our purpose was to find an inexpensive, yet reliable vapor sample container as an alternative to commercially available gasbags, which we found to be insufficient for our purposes either due to their large size or due to their degrading closure type. Therefore, we had inflated also other types of sample bags with pure, dry nitrogen gas. We found that only those combining laminated aluminum (Al) foil bodies and metal screw-lock valves yielded useful barriers against ambient vapor pressures (Fig. 3). These bags are mass products, originally produced for storing food or cosmetics and are much cheaper than existing diffusion-tight containers from specialized suppliers. All other tested combinations of bag materials and locks failed to prevent intrusion of ambient air which is

560 crucial for the storage of vapor samples. This corresponds to the finding of a previous study where Al-laminated bags also performed best in avoiding evaporation from soil samples (Gralher et al., 2021). Although the number of water molecules of these samples were three magnitudes higher than in our study we were not surprised to find Al-laminated bags again ranking best in material suitability.

Interestingly, not only the material but also the color of the material coating the diffusive barrier appeared to play a role. Clearly, different colors are a result of different chemical complex formulas used in the production and dyeing process. Unfortunately, they seem to come with different outgassing properties and spectral interference potentials regarding the intended isotope analyses. This issue always needs to be checked in advance. The presence of

certain organic compounds in a given gas sample may flaw laser-based isotope readings (Brand et al., 2009; Hendry et al., 2011), some of which can be identified by changes in the spectral parameter readings of the CRDS isotope analyzer.

In the "climate chamber" experiment, we actually sampled vapor in order to test how its concentration would change when the container is subjected to extreme moisture conditions

over a longer time period. Unexpectedly, we found that vapor concentrations slightly decreased over time (Fig. S1). This clearly contradicted the applied vapor concentration gradient relative to ambient conditions. Further, the extent of decrease was not affected by the canning of some of the containers as typically proposed for sampling water for dissolved gas analyses. Therefore, we argue that not only were the employed containers sufficiently

diffusion-tight against ambient meteorological forcings but also a fraction of the sampled vapor must have been absorbed by the inside coating of the containers (which had been repeatedly flushed with a dry atmosphere prior to vapor sampling). We can only speculate that mixing with previously absorbed vapor led to the observed, fairly uniform enrichment in heavy isotopes for both, $\delta^{18}O$ and $\delta^2H$. Presumably, the increase in vapor concentration inside

Al-laminated bags following pure nitrogen gas inflation in the first part of the material tests

(Fig. 3) was mainly due to release of previously absorbed vapor rather than via diffusive

intrusion from ambient.

**Field trial.** This "conditioning" effect from previous filling or exposures would also explain

why isotope data from the field-derived samples unanimously appeared to shift towards

ambient air in dual-isotope space (Fig. 4). This was likely because all bags used for this part

of our study had been freshly prepared. In doing so, during valve installation they had all been

exposed to the same ambient air (in the lab) with a homogenous vapor isotopic composition.

Apparently, that air was isotopically quite similar to the one recorded during the filling of the

field samples. The proportional, systematic shift of isotope values allowed for a robust

calibration scheme to be applied. This scheme used in situ and bag measurements of those

samples displaying extreme $\delta^2H$ values. It reproduced the intermediate in situ isotope values

representative for sampling of the bags with a precision sufficient for resolving naturally

occurring variations of meteoric waters. The observed discrepancies between in situ and bag

measurements indicate that the Principle of Identical Treatment needs to be followed also in

terms of ambient vapor isotopic composition when preparing a batch of sample bags for the

collection of unknown samples as well as co-measured standards. Additionally, we

recommend using bags of identical size and inflating them to the same volume in order to

maintain a uniform ratio of sample volume to internal wall area.

**Reusability of sampling bags.** Initially, we had used dry air or pure nitrogen gas in an

attempt to erase the signal of previous fillings and thus avoid carryover effects when reusing

the sampling bags. However, this procedure did not produce the desired outcome. Repeated

measurements from identically filled bags still displayed a clear isotope pattern of the

previous fillings (Fig. A2). The range of this pattern must be associated with the measurement

uncertainty if the latest filling had been an unknown sample and would thus prevent resolving

natural isotope variations. It was established before the first measurement after five days and then seemed to persist.

Therefore, we tried to find a different, easy-to-implement flushing or conditioning routine that would repeatedly enable precise vapor isotope measurements using our sampling bags. For this purpose, we systematically compared the impact of dry to moist conditioning on the
measurement precision and accuracy of vapor from isotopically diverse sources. From experience, we knew that room air vapor isotope signatures are usually quite persistent on shorter timescales as necessary for filling batches of sampling bags using high-flowrate devices such as vacuum pumps. Using this unlimited resource, we were able to establish a conditioning routine that is easy to reproduce and enables the precise reproduction of in situ
isotope values following impertinent calibration schemes (Table 3). In this context, the low SD of isotope readings from moist-conditioned bags proved to be a far better predictor of conditioning efficiency than the low vapor concentration of dry-conditioned bags. We therefore recommend conditioning entire batches of sample bags simultaneously with a moist, isotopically homogeneous atmosphere prior to each isotope sampling campaign.

Further, we emphasize that for any given vapor sample an isotopic shift must be expected between collection and subsequent, lab-based analysis. This shift became apparent e.g., during field application (Fig. 4) and the conditioning procedure (Fig. 6). It is characteristic and implicit to our method. However, it is proportional relative to the priming signal. Herewith, it becomes manageable through co-measuring calibration standards, prepared in
identically pre-treated bags.

Our conditioning routine was performed manually and its efficiency checked after every conditioning iteration. However, we are confident that it can be easily automated for future applications and its efficiency eventually assumed reliably without measurements as a matter of experience. Further, it seems that for the SD being a meaningful parameter for conditioning

efficiency, the conditioning time steps need to match the projected sample storage time. This also calls for automating the conditioning procedure.

Our aim was to develop a low cost solution as funding might not always be available and we wanted to avoid the high per-unit costs of commercially available gas sampling bags. Moreover, diffusion-tight bags from e.g. Analyt-MTC are made from plastic coated aluminum
foil (https://analyt-mtc.de/files/50/Produkte/4/Probenahme.pdf, in German), similar to the ones we used in our study. Therefore, similar adsorption issues must be expected due to the interior coating of the diffusive barrier. This means that before being readily available some kind of conditioning procedure would also have to be applied, even when re-using commercially available bags. Also, for any other sampling vessel a potential user will have to
verify the suitability in a way similar to the one we describe: in terms of diffusion tightness, contamination, as well as adsorptive disturbances from the inner layer of the vessel material itself.

So far, we are satisfied with the outcome of our tests and the achievable precision and accuracy. Nonetheless, we are aware that our method is not yet ready to go and further tests
are needed before unknown, field-collected samples can be processed. This refers not only to the issue of potential temperature discrepancies between samples and standards. Moreover, resulting differences in vapor concentration must be considered which likely induce additional challenges. The proportionality of the memory effect observed in this study, e.g. after applying the moist-air conditioning procedure was likely because we were able to
facilitate a consistent ratio of collected sample size and vapor reservoir absorbed to the bags' inner walls expressible e.g. in water vapor mole fractions. The effect of changes of this ratio has not been tested yet nor has a mathematical correction scheme accounting for this issue been found. Further, we are aware that the effort that needs to be put into assembling the necessary components as well as into pre-sample-collection tests and conditioning might still

have a deterring effect on potential users of our method. The same probably holds for the costly alternatives available in the market for collecting gas samples into flexible containers that would at least dispense with the assembly efforts. In this context, we admit that the option to use inexpensive off-the-shelf components as employed by e.g. Magh et al. (2022) is certainly appealing. Also, the option to reliably reproduce vapor isotope values after one or more weeks of storage remains a worthwhile goal.

On the other hand, we believe that the otherness of our method also holds several advantages relative to previous approaches for the collection and analysis of discrete vapor samples. In contrast to Havranek et al. (2020, 2022), the non-automation allows for maximum temporal flexibility regarding experimental designs. The spatial distribution of sampling sites is not limited by connectivity to a central sampling system and associated tubing lengths. As all employed components are small and light-weight, they can easily be deployed even in terrains that are rather difficult to access. In contrast to Magh et al. (2022), the achieved precision and accuracy (Table 3) allow for the resolution of even fine-scaled environmental isotope variations. Practically, our method does not require any tools, not to mention specialty tools during handling or refurbishing of crimp-sealed glass bottles and the lack of consumables such as septa makes it cost-efficient. The combination of the screw-lock mechanism and a sufficiently large sample volume is forgiving as it allows for multiple measurements from one sample bag. Also, flushing and re-filling are possible without wearing of the closing mechanism. Using evacuated, inflatable containers renders any in-field flushing obsolete thus reducing sampling time and increasing feasible temporal resolution. Further, it eliminates the risk of breaking glass during transport and handling. Finally, our method does not require additional gas sources during analysis and allows for directly interpretable measurement readings without the need to identify the relevant isotope data section via derivation of the vapor concentration readings.

 **5 Conclusions**

We present a new method for the collection of discrete water vapor samples in the field and subsequent storage and isotope analysis in the lab. After systematic material testing, we identified the optimum combination of inflatable bag and closure type that guarantees air-tightness and avoids sample contamination by material outgassing. Our custom-made method uses off-the-shelf components only and is easy to use, cost-efficient, sustainable and allows for multiple measurements. Further, it allows for direct interpretation of the obtained isotope results. The achieved precision and accuracy are not only suitable for labelling experiments but also sufficient for resolving natural variations of water stable isotope signatures. Preparation and co-measurement of calibration standards are indispensable for our approach in order to correct for the implicit shift of the obtained isotope signal, induced by the mandatory conditioning procedure. We are convinced that the conditioning procedure can be automated, which would further reduce the per-sample workload when re-using sampling bags. The presented approach allows for collecting vapor samples from soil matrix and plant tissues in remote settings without an isotope analyzer in the field. The method therefore widens the applicability of minimally invasive in situ approaches of matrix-bound water stable isotope observations. We are therefore confident that our method will open new observatory paths and thus contribute to novel insights in hydrology, soil science, plant physiology and related disciplines. Future tests will have to find ways of dealing with temperature and vapor concentration discrepancies among samples during collection and relative to co-prepared standards.

## 6 Appendix A: Protocol for identifying appropriate bags

The procedure of testing and the requirements that had to be fulfilled in our study by the tested gas sampling bags are summarized in the protocol. Any other material or commercially available bags can also be evaluated by passing this protocol.

Table A1: Suggested test protocol.

| | Properties tested | Filling bags with: | Aim | Observed instrument variable *L2120-*i** **L2130-*i* | Target value for passing test |
|---|---|---|---|---|---|
| Test 1 | Diffusivity | dry N$_2$ | Identify mixing with ambient air | Water content: 'H2O' spectral gas matrix variable: 'h2o_y_eff_a'* 'h2o_vy'** | < 1000 after 2 weeks  ~1.03* ~0.46** |
| | Spectral interference | | Identify substances spectrally interfering with measurements, thus flawing isotope readings | Water content: 'H2O' Stable isotopes: 'd18O' 'd2H' spectral gas matrix variable: 'h2o_y_eff_a'* 'h2o_vy'** organic contamination: 'MeOHampl'* 'CH4_conc'** | < 1000 after 2 weeks     ~1.03* ~0.46**  ~0.00095* ~-0.00012** |
| Test 2 | Reliability  Only conducted when tests for diffusivity and spectral interference were passed | Vapor from 3 different sources with known isotopic composition | Evaluate isotope measurement in respective bag type by calibrating intermediate validation standard with known and observed values of two calibration standards (heavy, light) | Water content: 'H2O' Stable isotopes: 'd18O' 'd2H' | similar for all 3 standards  Calibrated values should match known values with deviations < 0.4 ('d18O') and < 2.0 ('d2H') or better |

| | | | | | |
|---|---|---|---|---|---|
| **Test 3 (optional)** | Conditioning for re-use of bags | Room air (repeatedly) | Homogenize memory effect across entire sampling bag batch to make it manageable via pertinent calibration schemes | Water content: 'H2O' Stable isotopes: 'd18O' 'd2H' | ~5000-20000 SD of whole batch of bags << 0.4 ('d18O') << 2.0 ('d2H') (see Fig. 5) |

## Appendix B: Memory effects in re-used bags.

In re-used bags, we observed isotope data consistently trending towards the values recorded previously from the respective sample bags, although evacuated and flushed three times with dry $N_2$. These 'memory effects' were proportional to the differences between the current and the previous bag measurement (Fig. A1).

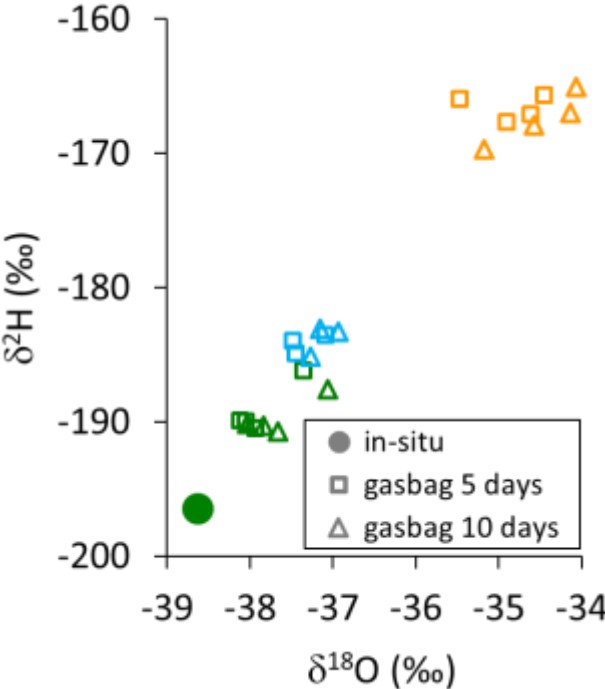

Figure A1. Dual isotope plot of gasbag measurements of re-used bags 5 days (open squares) and 10 days (open triangles) after all of them being filled with a constant flowrate of 150 mL/min via a WIP from one single isotopic reservoir (filled dot). The different colors indicate the differing isotopic levels of the previous samples, stored in the respective bags.

## Author contribution

BH, SS, MR and MW designed the first part (material selection, field trial) of the experiments, BH and SS carried them out. BH and BG designed the second part (climate chamber, conditioning procedure) of the experiments and carried them out. BH, BG and MW analysed and interpreted the data. BH and BG prepared the manuscript with contributions from all co-authors.

## Acknowledgments

The authors thank Janine Heitzmann for her tireless commitment in the lab during material testing. This work was partly funded by the priority program SPP 1685 "Ecosystem Nutrition: Forest Strategies for limited Phosphorus Resources" of the German Research Foundation (DFG; contract numbers: GE 1090/10-1 and WE 4598/7-2) and the University of Freiburg through the Open Access Publishing funding program.

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
