# Peer review of "Technical note: Discrete in situ vapor sampling for subsequent lab-based water stable isotope analysis"

_Hydrology and Earth System Sciences, 2022_

## Author Comment (AC1)

**RC1**: 'Comment on hess-2022-393', Anonymous Referee #1, 05 Jan 2023 reply

This manuscript presents a method to store water vapor sampled from matrix-bound waters in the field for later stable isotope analysis in the lab by laser spectrometer. I think it is a good contribution and such eventually be published, though I cannot give my endorsement without the following general and specific comments being addressed.

We thank the reviewer for the constructive comments, which will help to improve the manuscript.

General comments:

The procedure and calculations used to calibrate the observed vapor measurements and assess their accuracy and precision are vague at best, and opaque at worst. This is troubling, and almost suspicious, because these metrics are the backbone of how to assess the effectiveness of the proposed methods. Why would the authors be so vague about such a fundamental component of assessment? I cannot endorse this manuscript without seeing a much more rigorous presentation of how they assessed accuracy and precision. Except for one instance, they don't even bother to state what the $\delta^{18}O$ and $\delta^2H$ values of the test waters they used were.

We will add more details to the description of our calibration procedure to make clearer how we assessed precision and accuracy of the obtained isotope values.

Also missing is a simple section that presents their recommendations of how to best employ the method. I suggest at the end of the discussion they present a brief and easily digestible "Best Practices" section. Instead, what is now included is a litany of what worked better than others. This leaves the reader thinking, "OK, so what do I do? Just tell me and I will do it."

We will add a detailed SOP/ best practice to the revised manuscript for potential users.

Specific comments:

P1, L1: The use of "mobile" in the title is somewhat misleading. It implies that the analytical system is mobile, which is not the case. Of course, any sampling is inherently mobile, as you have to move to the location to do it.

Our sampling system is mobile in opposite to the immobile in situ setups, which we try to develop further. There, a field site is instrumented with probes and analyser, with the disadvantage that the sampling site has to be chosen according to the setup's restrictions and requirements (slopes, tubing lengths, distances to the instrument, access to power supply, etc.). Our method allows the position of sampling to be detached from the position of the analyzer which means the sampling is mobile relative the lab-based measurement.

Also, why "discrete"? Isn't any sample discrete? I suggest rephrasing the title to better describe what you are doing here: A method for sampling in situ water vapor to make stable isotope measurements. Or as you state later (P5, L12): "the aim of this study is to develop a technique to collect discrete vapor samples in
the field for subsequent lab-based analyses"

We sampled 'discrete' in contrast to the 'continuous', on-site vapor measurements, the capabilities of which we want to expand with our approach. We will consider changing the title under consideration of the reviewer's suggestion.

P2, L15: Dawson and Ehleringer, 1991 is certainly a pioneering paper, but they didn't actually identify the water source for their streamside trees, just that the trees were not using soil water. Suggest including Oerter et al, 2017 because they did actually find the water source (using water vapor isotopes): soil water.

> Oerter, E, G Siebert, D Bowling, G Bowen, 2019, Soil water vapor isotopes identify missing water source for streamside trees, Ecohydrology, v. 12: e2083, https://doi.org/10.1002/eco.2083

The suggested additional literature will be included in the revised manuscript.

P5, L11: Either provide direct evidence and citation for your claim of "clearly biases the isotope data", or leave it out. Otherwise this is an unsubstantiated claim.

To eliminate the confusion, we will rephrase this section to: "A sophisticated calculation procedure is necessary for both approaches (Havranek et al., 2020; Magh et al., 2022) to remove the effects of the initial pulse of water vapor during the start of the measurement phase. This initial pulse is mixed with pre-sample vapor which clearly biases the obtained isotope data."

P7, L8-20: I don't understand this section completely. Are you doing this in the lab or field?

> The effect of changing gas throughflow rates on the isotopic composition was investigated in a lab experiment. We will add this information to the revised manuscript.

> Why is the analyzer running at the same time you are filling sampling bags? Why is there excess gas flow if you are sucking moist air out of a bag and into the analyzer?

> No bags were filled in this part of the study. In this section we investigate the effect of changing gas throughflow rates on the isotopic composition. We increased the throughflow rates through the probe and thus produced more vapor than the analyser takes. Therefore, a spillover was needed.
> We tested the effect of increased flowrates in order to facilitate shorter bag filling times in the future application.

> If you are in the field, how do you power the mass flow controller and pump?

> In a field application mass flow meter and pumps can be powered by e.g. 12 V batteries.

P9, L1: Suggest including a figure with pictures of the bags, focused on the fittings. These are hard to visualize otherwise.

As suggested, a figure with pictures of the bags will be added to the supplement.

P9, L9: What measurement or spectral parameter would identify diffusion or outgassing?

OK, I see later that you address this. I suggest either briefly discussing or pointing the reader to where you address it later.

The spectral parameters are introduced in the method section on p. 7. There, we will add which parameter is indicative for what.

P9, L19: Seems like you could employ a better, more intuitive bag naming system, so that the reader could reasonably understand what bag/valve combination you are talking about, rather than having to look it up in Table 1. I see you have made efforts toward this, but it could still be improved. Why does it matter if they are silver or red or gold? Is the material itself different? To replicate your results and use your method does the reader have to get red bags?

OK, I see later that you address this. I suggest either briefly discussing or pointing the reader to where you address it later.

We were surprised ourselves to find differently colored bags to perform differently, as we describe in the result section and discuss later on.

P10, L13: Bags from inside the storage cans?

We meant all the bags from inside the box. For clarification, we will add that we mean those from inside the storage cans as well as the uncanned ones.

L10, P18: Please define and describe DDS mode.

The working principle of the probe in the different modes (e.g. DDS) is described by Volkmann et al. (2014). We will describe the working principle in the revised manuscript but omit the abbreviation used in the cited literature to eliminate the confusion as other working principles were not used in this study.

P11, L4: Were the bags analyzed under the same conditions as the wet-sand boxes were kept and the bags filled in? Or taken back to the lab and analyzed there?

In this field experiment everything was carried out under identical conditions in the field. Also the bags were analysed with the analyser in the field two hours later.

P11, L5: I don't understand this calibration procedure. What was the d2H and d18O values of the water used for the wet-sand boxes? Why use the extreme d2H values? What about d18O? This point is extremely important! The assessment of your whole study hangs on how you calibrated, and how you assessed the accuracy and precision of measurements derived from your method.

By writing 'the extreme d2H values' we describe which two of the four reservoirs we considered as calibration standards and which were thus used as validation standards in this field experiment. This selection was maintained for the calibration of the d18O values as well. We will rephrase the respective section for clarification.

P12, L7: What were the d values of the "isotopically diverse" sources?

We will elaborate on the d values of the isotopically different sources in the revised manuscript.

P12, L11: This explanation of the precision and accuracy is very vague. What were the "calibrated isotope reading" upon which the SD was calced and how many? What is the "respective target value"? This is fishy, because this is the metric by which your whole method must be evaluated by.

The 'respective target value' is the in situ value we measured in the lab which we tried to reproduce with our calibration procedure.

P12, L15: This seems to me to be due to the concentration-dependence of the d2H and d18O values. This is well known for Picarro L-2130 generation instruments operating in continuous flow mode. The magnitude of the effect is larger than I have experienced though, so it could have multiple causes.

OK, I see later that you address this. I suggest either briefly discussing or pointing the reader to where you address it later.

We addressed this matter on P12 in L17 and also later in the discussion.

P13, L8: Finally, you tell us what the d values of the water you used were. This information needs to be included throughout.

Thanks for pointing to this; we will add the information to the method section.

P14, L9: This approach where you briefly describe the bags, then mention their name code helps me understand which bags you are talking about. I suggest you do this throughout the entire manuscript.

We will add the name codes where they are missing.

P14, L15: Here again with no bag type description I don't know which bags you are talking about. Use the description approach in the previous paragraph to help your reader understand which bags you are talking about.

We will expand this introductory sentence with the requested bag type description.

Figure 6: It is hard to tell the various symbols apart, especially since so many overlap. I suggest naming the symbols larger and more distinct colors. This applies to the other figures as well.

We found that the figure was better readable when using color gradients rather than different colors or symbols (as e.g. in the supplement Fig. S3 and Fig. S4). We will try to improve the figures by enlarging the symbols.

P22, L13: OK, now I see that you are considering vapor-concentration effects. This discussion section is nice and complete.

Thank you.

P 23, L 7: OK, now I see you are discussing the color issue.

We were surprised ourselves to see that differently colored bags performed differently.

P 26, L22: I suggest a short section here that summarizes "Best Practices" about how to employ your method.

We will include a detailed SOP/ best practice for potential users.

---

## Author Comment (AC2)

**RC2**: 'Comment on hess-2022-393', Anonymous Referee #2, 06 Mar 2023 reply

Thank you for letting me review the manuscript "Mobile, discrete in situ vapor sampling for measurements of matrix-bound water stable isotopes". The authors present an alternative way (in comparison to Magh et al., 2022 and Havranek, 2021) of obtaining water vapor samples for analyzing the isotopic composition of soil and xylem water samples obtained with gas-permeable membranes and storing them in gas-tight sampling bags. In their study, the authors test multiple aspects related to sampling, storing and analyzing water vapor in different types of sampling bags. The authors conclude that sampling bags have potential to be used as sampling vessel for water isotope samples and have multiple advantages over bottle-based previous approaches.

We thank the reviewer for the valuable, constructive comments and for addressing many important aspects, which will help to improve the manuscript.

The manuscript is interesting and the efforts of making in situ methods more feasible, e.g., not having to carry an isotope analyzer to the field, is very much appreciable. However, the whole procedure of preparing, flushing, conditioning the bags appears extremely laborious and error-prone – it would've been appreciable if at least one commercially available, ready-to-go product would've been tested. The reported precision is not a true precision, as the isotope values are calibrated with true in situ measurements (which the applicant of the method would not have).

Thank you for raising this point. What we describe in our manuscript can be seen as the preparation and measurement of calibration and validation vapor standards for processing of unknown vapor samples. Such standards are indispensable and thus need to be prepared by the applicant of our method. In any case, the true values of these standards must be known. Therefore, we argue that the reported precision can be considered the true precision.
It might be a misunderstanding: the standards don't need to be prepared in the field, they could be prepared and analysed in the lab.

In the discussion section, I got the impression that the authors try to convince the reader 'bags are better than bottles', rather than highlighting the advantages and disadvantages objectively.

We will rework the discussion to be more objective.

Having worked with both methods, I also would've liked to see different storage temperatures and/or shifts of temperatures being tested (e.g., should all samples be heated in order to avoid condensation-related effects?).

Regarding potential saturation effects we see the selected high flowrates through the probes as an advantage. That way we produce vapor that is far from saturation. Therefore, heating or dilution of the samples to avoid condensation is not necessary.

The manuscript definitely has potential to be published, but it is unfortunate that several interesting aspects (e.g., isotopic equilibrium, recommended flow-rate, protocol for potential users, temperatures) were not investigated. In my opinion, it should be at least stated that further tests are required and the method is not yet ready-to-go.

Different temperatures during storage or sampling (which would in turn cause different vapor concentrations) were not in the scope of this study. Here, we focused on the reproducibility of bag-filling data. Due to the increased flowrates (for short sampling times in the field), the

water vapor content was below saturation anyway. Thus, condensation-related effects were not an issue.

But we absolutely agree that temperature issues are of importance and need to be investigated prior to field-readiness. We will add a statement on this in the revised manuscript.

Also, a SOP (protocol for potential users) will be added.

Hence, I recommend revision and reconsideration.

Main comments:

- Please use continuous line numbering in the future (not starting with line 1 every page)

Thanks, we will do so with our next manuscript.

- Section 2.1.: This chapter is testing the flow rates through the WIP's and the reader gets the impression those can be chosen arbitrarily; but this is not true. For both soils and xylem, equilibrium fractionation inside of the WIP is required.

We argue that identical conditions need to be fulfilled which we did. Our chosen flowrate was identical for all standards and samples. Given the reproducibility of isotope values from non-equilibrium vapor samples in our study, we think that flow rates through identically dimensioned WIPs do not need to facilitate isotope equilibrium. However, this would be an issue requiring extra attention when dealing with e.g. the stem borehole method on trees with heterogeneous diameters.

- If flow rates are too high, this requirement cannot be fulfilled anymore. Marshall et al. (2020) provided a way to calculate the maximum flow rate possible with their stem borehole method. However, this seems to not have been tested for WIP's, where the membrane might have an influence on the exchange times. If this was tested, it would be great to cite that or at least provide information on this issue.

This matter was tested by Volkmann et al. (2014, doi: 10.5194/hess-18-1819-2014). They optimized the probe dimensions and contact area of the porous tip of the probe to the flowrate demanded by the isotope analyzer. As they aimed at developing an equilibrium-based in situ method, they didn't test other flowrates than the one given by the instrument. We will add this information to the revised manuscript.

We are aware that we were beyond equilibrium in this study, but aimed at shortening the filling times in the field. We therefore additionally investigated the effect of increasing flowrates and were able to reproduce the intermediate values (validation standards).

- Otherwise, the potential applicant of the method is at risk on obtaining false data (e.g., when trying to fill the bags as fast as possible). While this aspect is briefly touched on later in the manuscript, this key issue should imo be of utmost importance and needs to be elaborated thoroughly. While equal treatment principle might be able to correct the offset, it should still be the goal to limit post-corrections, i.e., to obtain a sample under equilibrium isotope conditions.

We agree that equilibrium conditions would be nice to have. However we argue that necessary post corrections are identical in our study when applying identical flowrates. We will emphasize this aspect more strongly in the revised manuscript.

We also agree that post-corrections should be limited, but in the range of our study, i.e. in the case of constant gas flow and pumping rates as well as similar climatic conditions, our calibration procedure is the only post-correction necessary – and it additionally includes the correction due to the conditioning. Mathematically, it is identical to pertinent calibration schemes with no extra calculation steps necessary due to non-equilibrium conditions.

- While it is a great effort to test sampling bags with a focus on relatively cheap material, all of the tested bags are somehow custom-made bags where the potential user needs to glue-seal, change, glue something or put silicone. Imo, at least one commercial product (see my comment later, there are definitely gas sampling bags with metal/PTFE valves of the volumes 500ml/1L and others available) should've been tested, both as reference but also as an option for applicants who might prefer a readily available sampling vessel.

Thank you for raising this point. We agree that due to the necessary preparation steps the method gets more error prone.
We are aware of commercially available products, which according to our query range from €20 - €40 per bag which we find prohibitive for single use in large-scale applications. Two commercially available Linde plastigas bags were tested in our study, but failed to provide trustworthy isotope data due to contamination issues likely caused by outgassing of the bag material.
Our aim was to develop a low cost solution as funding might not always be available and we wanted to avoid such high per-unit costs. Moreover, diffusion tight bags from e.g. Analyt-MTC are made from plastic coated aluminum foil (https://analyt-mtc.de/files/50/Produkte/4/Probenahme.pdf, in German), similar to the ones we used in our study. Therefore, similar adsorption issues must be expected due to the interior coating of the diffusive barrier. This means that before being readily available some kind of conditioning procedure would also have to be applied, especially when re-using commercially available bags. Also, for any other sampling vessel a potential user will have to verify the suitability in a way similar to the one we describe: in terms of diffusion tightness, contamination, as well as adsorptive disturbances from the inner layer of the vessel material itself.

- (This is related to main comment 1, sorry for being repetitive but it is an important aspect) Section 3.1: This section leaves me puzzled. It is reported here that there was a kinetic fractionation effect observed at all flow rates. But the foundation of vapor-equilibration methods is to have no kinetic fractionation in place, i.e., only equilibrium fractionation in order to derive the liquid water isotope values. If this assumption is not fulfilled, how does one obtain a reliable isotope value with this method? Is it calibrated? I do not think that equal treatment of isotope standards and samples can be applied here, because the isotope standards are taken from liquid water (with a much greater water contact surface) and not from a matrix such as soils or xylem.

In this study, we tested the necessity of isotope equilibrium for water vapor sampling. We found that also under non-equilibrium conditions, in-situ isotope values could be reproduced with a very good precision and accuracy. This expands the findings of previous studies which aimed at facilitating equilibrium conditions when sampling water vapor from soils or stem boreholes.

There must be a misunderstanding regarding liquid water being the source of our vapor samples. We used re-wetted soils, not pure liquid water. By measuring both, calibration and validation standards with the WIPs fractionation effects of the membrane were eliminated.

- 16. l 23-24 and Fig.4: I am not sure how useful the calibration shown is for practical applications. It is good to show for a methodological test that values can be calibrated to match the in situ measurements, but for real measurements one would need to measure in situ in order to correct the bag values – which is contrary to the goal of the manuscript.

The goal of this study was not to avoid standard measurements which are indispensable for any kind of isotope measurements. Instead, what we describe in our manuscript can be seen as the lab-based preparation and measurement of calibration and validation vapor standards for processing of unknown vapor samples collected in the field. Provided a proper correction scheme, it is foreseen that it won't be necessary to measure in situ values in the field. For calibration, bag measurements of the standards are referenced against the respective in situ-measurements both of which are performed in the lab. It is crucial, that the conditioning procedure has to be identical for standards- and sample-bags. Of course, future correction schemes also have to consider temperature-related effects causing e.g. variable vapor concentrations among samples and relative to the lab-prepared standards. However, this will be subject to future studies.

- It would be more useful, imo, to honestly report the deviations of the bag method from the in situ measurements rather than the 'calibrated' values (or at least report both).

We will additionally report the 'raw' values from the bag method for comparison, but accuracy and precision of repeated validation measurements after calibration is in our opinion the quality measure to compare our method to other methods.

Abstract:

l.24/25: were soils and xylem tested? Or the validation procedure only performed with isotope standards

We tested two different types of soils. The validation standards were re-wetted sand, not liquid water.

l. 25: I suggest rephrasing the last sentence. One could also obtain time-series with destructive methods if there is no analyzer on site. Imo the real benefit here is that the method is minimal-invasive and the tree is not riddled by an increment corer when doing repeated measurements.

Will be changed as suggested.

Manuscript:

p.2 l.7-18: I know these classic citations are popular and correct, but I feel like it would really be good to put at least a couple of recent studies here.

Will be added as suggested.

p.3l.1: why are laser instruments not mentioned as conventional method? Regardless of the organic contamination issue they are used widely on extracted samples

We agree, we will rephrase accordingly.

p.3. l.6-12: Imo it should be said here that 1) taking many xylem samples from a tree can kill a tree or give fungi access to the tree; 2)Taking branches can be challenging and even impossible for tall trees; and 3) that it is impossible to take samples at the exact same position when sampling destructively.

We will include these points to the revised manuscript as suggested.

p.4. l.7-8: correct would be 'stem borehole method' , perhaps consider to citing Kühnhammer, K., Dahlmann, A., Iraheta, A., Gerchow, M., Birkel, C., Marshall, J. D. and Beyer, M.: Continuous in situ measurements of water stable isotopes in soils, tree trunk and root xylem: field approval, Rapid Commun. Mass Spectrom., e9232, doi:10.1002/RCM.9232, 2021.

Sorry for the mistake, will be corrected.

p.5 l. 11: 'clearly biases' I recommend weakening the wording, e.g., "might bias" – where is the proof that it clearly biases anything?

We meant to say that the initial pulse from the sample reaching the analyser is flawed by mixing from previous sources, e.g. ambient air. We did not intend to say that the calculation procedure results in biases. To eliminate the confusion, we will rephrase this section to: "A sophisticated calculation procedure is necessary for both approaches (Havranek et al., 2020; Magh et al., 2022) to remove the effects of the initial pulse of water vapor during the start of the measurement phase. This initial pulse is mixed with pre-sample vapor which clearly biases the obtained isotope data."

p.5. l.12: Therefore,

Will be change as suggested

p.8l.2: This is not true; there are definitely gas sampling bags with valve systems on the market with 250ml, 500ml or 1L volumes (e.g., Analyt-MTC)

We will rephrase this section to include other commercially available gas sampling bags which we did not test further for reasons stated above.

chapter 2.2.2: The first sentence should state what this experiment was set up for. It is currently not very understandable what was actually done here. I guess the idea was to test if relative humidity outside of the bags has an influence on the exchange with the bag inside? When speaking of 'Meteorological forces', I was also expecting an experiment with different storage temperatures, which could have a tremendous effect on the vapor sample (e.g., condensation). Was this assessed somehow?

We will rephrase the sentence to be more understandable. Yes, the idea was to test if a high gradient in relative humidity had any measurable impact on the stored sample. However, temperature-related effects were outside the scope of this study.

p.10 l. 15: the projected

Will be changed as suggested

p.10.l.22 – p.11. l.9: Imo it would've still be necessary to convert and compare the values obtained by WIP's and the bags with the values for the isotope standards. In fact, this experiment could've been used for evaluating the effect of the flow rate on the isotope values obtained by the WIP's in general, which I personally find crucial. If this was done before, it should be cited.

In the first place, this study is about collecting vapor samples and determining their vapor isotope values. At a later stage, such vapor samples can be used to characterize the liquid water sources they have been obtained from. However, this would require additional tests accounting for temperature-related issues which was outside the scope of this study.

The setup investigating the effect of the flow rate on the isotope values obtained by the WIPs has been described in section 2.1.

p.11 l. 24: "further improve reusability" - strange wording, suggest "We tested reusability of the bags…"

Will be rephrased.

p.13. l. 11: Why is 150 ml/min the target flow rate? This is the rate with the highest effect of kinetic fractionation on the sampled air, according to Fig. 2.?

The selected high flow rate was a trade-off between shorter filling times (more samples per collection day) and sufficiently high water vapor concentration. We aimed at filling times of ~5 minutes and sample volumes of ~750 mL, which corresponds to 150 mL/min, and tested the flowrate effects from equilibrium (~35 mL/min) through 150 mL/min. Presumably, flowrates exceeding 150 mL/min would have led to even higher fractionation effects as well as lower vapor concentrations.

p.15 l.7: L2130i analyzer – before it is mentioned a 2120i was used…typo?

Depending on the availability of the instruments, we used the L2120-*i* as well as the L2130-*i* analyser in different parts of this study.

p.15. l.14: The apparently tested commercially available plastigas bags are not mentioned in the methods section (Table) and a comparison with those comes out of nowhere here – please add/move to methods or take out completely. I suggest opting for the latter as it does not add substantial information.

True, they weren't in the table as they failed very early in our tests, sorry. We will add those commercially available bags to the method section.

p.16 l.8: quite … as intended

The sentence will be rephrased to "Temperature was quite stable inside the climate chamber as intended. It ranged between 18.1°C and 16.3°C." in the revised manuscript

p.17 l. 10-13: What are the authors referring to when speaking of "the values recorded previously from the respective sample bags"? Is it referring to a memory effect? The first two

sentences of this section are very hard to understand in general. Please clarify/rephrase/extend explanations.

Yes, this section is about erasing potential memory effects to enable re-use of the bags and thus reduce per-sample cost and effort that had to be put into constructing the bags. We will rephrase the section for clarification.

Chapter 3.4.2 & Figure 6: see my main comment on the 'calibrated' values. If in situ measurements and bag measurements deviate, this should be reported rather than calibrated with an approach that cannot be transferred to other field studies.

In our study, in situ measurements and bags measurements do deviate. This is mainly due to the applied conditioning routine and we have no problem with reporting these deviations. We found that they are very systematic, shifting bag measurement proportionally towards conditioning values.

Given this proportionality, we argue that such deviations are generally dealt with by the application of pertinent calibration schemes that would also eliminate deviations between the isotope values of VSMOW-referenced liquid water in-house standards and the raw values of their measurements in routine liquid water analyses.

p.20 l. 21-25: How was the conclusion drawn that 12.000 ppm @ 21°C would be sufficient to "enable sufficiently precise isotope measurements for resolving natural variations" (citation missing)? It simply means the analyzer can still measure the sample – it does not tell, however, if isotopic equilibrium was reached in the probe. One could calculate the ppm value for saturation @21°C and compare it to the measured 12.000 ppm – if there is a large difference, there was no isotopic equilibrium reached and the measured isotope values would be significantly off. (Might be calibrated via equal treatment, but see comments above on this). At minimum, these issues should be reported and commented.

Here, we refer to the fact that laser-based isotope measurements, e.g. on a Picarro analyser, are subject to higher uncertainty (noise) when performed on low-concentration samples. We refer to the manufacturer's data sheet for the feasible measurement precision at the obtained vapor concentration. We find this precision to be sufficient to resolve naturally occurring isotope variations which, e.g. in precipitation, are usually two orders of magnitude higher.

We agree that there is no way of identifying equilibrium conditions from vapor concentration alone.

p.22. l.4-17: It is true what is written here, but the first and foremost goal of such methodological developments should be to obtain the true isotope value for whatever medium studied. The authors report that that "neither the previous recommended nor our newly selected settings result in complete isotopic equilibrium". This is important – and imo it would've been important to work on improving this aspect first. The analyzer draws approximately 30-35 ml/min, so much lower flowrates could've been tested in order to obtain isotopic equilibrium. As per now, the reader is left without a clear message – if the proposal is to stick to higher flow rates, how can one assure accurate isotope data?

We meant to explain that in our case isotope equilibrium conditions are not necessary to obtain sufficiently precise and accurate isotope measurements of water vapor validation standards

p.24. l.11-12: but only if true in situ measurements exist!

As stated above, what we describe in our manuscript can be seen as the lab-based preparation and measurement of calibration and validation vapor standards for processing of unknown vapor samples collected in the field. This means that like for any liquid water isotope analysis the true value of the employed calibration and validation vapor standards must be known.

p.24 l. 15-16: How would the practical setup for equal treatment look like? The standards would probably sampled from liquid water, whereas the samples would originate from a matrix (soil or xylem). It would be helpful for the reader to elaborate on this imo.

The standards we used were sampled from re-wetted sand, not liquid water. From such moist sand standards the in-situ measurements will be obtained in the lab, with probes identical to the ones used in the field (soil, xylem). We will rephrase to make it clearer.

P.26 l.4-22: A more objective discussion would help the readers. There are also a number of disadvantages using bags, e.g., they are much more vulnerable when being pressed, they require more space compared to previous tests using bottles, they need to be prepared (putting the septum, using silicone-glue which ultimately can affect isotope values) manually. Also, bottles do not require any pre-treatment which is laborious and time-consuming. As stated before, the reported precision in the study is obtained by calibrating the bag results with in situ measurements which are not available in the field if one really wants to not take the analyzer to the field. Having worked with both the bag sample collection method and bottle approaches, using bags has been tremendously more difficult compared to bottles. This might be a personal preference, but I think a more objective discussion in general (rather than 'convincing' the reader) would benefit the study.

We will rework the discussion to be more objective.

Figure 2: While the information is interesting, the reader is left without an explanation what is the takeout of this graph. It would help to put this in context – e.g., which gas flow rate is recommended?

Will be added as suggested.

As the outcome of the paper, I was expecting a recommendation for potential users, and this is not contained at present. It appears that the complete procedure requires many steps and this makes the method error-prone. I think the manuscript would benefit a section on practical issues.

We will include a detailed SOP/ best practice for potential users to the revised manuscript.

---

## Referee Report (RR1)

The authors did a good job in revising this manuscript and it reads much smoother now. It will be a useful contribution and potentially valuable new method to be considered. However, I still disagree with the claim of the authors that using high flow rates for the sampling are an advantage and their calibration of isotope data for obtaining soil and xylem isotope values is following the principle of equal treatment. (I attached my original questions and author responses down below).

When using high flow rates, no full equilibration inside of the membrane is reached anymore. This membrane has a certain length and exchange area with the tree xylem or soil, and it is true that by changing the membrane length, the chance for equilibrium conditions, a prerequisite for all direct-vapor-equilibration methods in the past, can be reached. The authors claim that they do not need equilibrium conditions in their membrane, because standards are treated the same way and values corrected thereafter. But this argumentation has a major flaw: Yes, same flow rates are used for standards and samples, but in both xylem and soils the membrane sits in a porous media with a limited maximum water content (50% are rarely exceeded). In contrast, vapor exchange in the liquid water standard (or its headspace) will be much faster and isotopic equilibrium reached with much higher flow rates. As a result, semi-equilibrated vapor samples (from xylem or soil) are calibrated with fully or almost-fully equilibrated isotope standards; hence, the kinetic isotope effect on standard and sample is different. And this is why I do not agree that equal treatment principles apply here and suggest initially that this should've been tested. If one key principle of direct-vapor-equilibration methods - isotopic equilibrium - is violated, there needs to be a solid proof-of-concept in my opinion (and reviewer #1 also raised a similar concern in his initial comments). The revised manuscript does still not clearly address this sufficiently in my opinion. The authors mention the validation standards, but these also don't resolve the abovementioned concerns (because those are also based on exchange with liquid water).
In the revised manuscript, the authors also say that the good repeatability of their measurements is a proof of the method; however, a good repeatability has nothing to do with obtaining the correct isotope value. On a side note, and because the authors mention the method: the chance for isotopic equilibrium using the stem borehole method is much higher because no constraining membrane is used and the contact area with the matrix is much greater. Also, equilibrium conditions are tested by determining relative humidity of the sample after passage through the borehole. I simply do not understand why such a test was never performed with the WIP's (other than Volkmann & Weiler 2014) and is not regularly done when taking vapor samples.

Q: Section 2.1.: This chapter is testing the flow rates through the WIP's and the reader
gets the impression those can be chosen arbitrarily; but this is not true. For both soils
and xylem, equilibrium fractionation inside of the WIP is required.
A: We argue that identical conditions need to be fulfilled which we did. Our chosen flowrate was identical for all standards and samples. Given the reproducibility of isotope values from nonequilibrium vapor samples in our study, we think that flow rates through identically dimensioned WIPs do not need to facilitate isotope equilibrium. However, this would be an
issue requiring extra attention when dealing with e.g. the stem borehole method on trees with heterogeneous diameters. We addressed this issue in the revised discussion.

Q: If flow rates are too high, this requirement cannot be fulfilled anymore. Marshall et al.

(2020) provided a way to calculate the maximum flow rate possible with their stem borehole method. However, this seems to not have been tested for WIP's, where the membrane might have an influence on the exchange times. If this was tested, it would be great to cite that or at least provide information on this issue.

A: This matter was tested by Volkmann et al. (2014, doi: 10.5194/hess-18-1819-2014). They optimized the probe dimensions and contact area of the porous tip of the probe to the flowrate demanded by the isotope analyzer. As they aimed at developing an equilibrium-based in situ method, they didn't test other flowrates than the one given by the instrument. We added this information to the revised manuscript. We are aware that we were beyond equilibrium in this study, but aimed at shortening the filling times in the field. We therefore additionally investigated the effect of increasing flowrates and were able to reproduce the intermediate values (validation standards).

---

## Author Response (AR2)

The authors did a good job in revising this manuscript and it reads much smoother now. It will be a useful contribution and potentially valuable new method to be considered. However, I still disagree with the claim of the authors that using high flow rates for the sampling are an advantage and their calibration of isotope data for obtaining soil and xylem isotope values is following the principle of equal treatment. (I attached my original questions and author responses down below).

We thank the reviewer for their favorable evaluation. We also thank the reviewer for being persistent on the important issues of flow rate and the principle of identical treatment. In the revised manuscript we made clearer that the high flowrate was applied to facilitate shorter filling times. While we consider this suitable within the scope of our study, we did not mean to make a general statement devaluating other, equilibrium-based methods. We rephrased the respective sentences.

Regarding the principle of identical treatment, we do think that the calibration and validation standards we prepared in our lab-based experiments were treated identical. Importantly, this refers to gas flow rate, matrix potential and temperature (and thus vapor concentrations). We admit that while the former is easy to control the latter two variables will inevitably change among samples and relative to the standards in the foreseen field application. Certainly, this needs extra attention prior to our method being field-ready. We regret missing this point earlier.

When using high flow rates, no full equilibration inside of the membrane is reached anymore. This membrane has a certain length and exchange area with the tree xylem or soil, and it is true that by changing the membrane length, the chance for equilibrium conditions, a prerequisite for all direct-vapor-equilibration methods in the past, can be reached. The authors claim that they do not need equilibrium conditions in their membrane, because standards are treated the same way and values corrected thereafter. But this argumentation has a major flaw: Yes, same flow rates are used for standards and samples, but in both xylem and soils the membrane sits in a porous media with a limited maximum water content (50% are rarely exceeded). In contrast, vapor exchange in the liquid water standard (or ist headspace) will be much faster and isotopic equilibrium reached with much higher flow rates. As a result, semi-equilibrated vapor samples (from xylem or soil) are calibrated with fully or almost-fully equilibrated isotope standards; hence, the kinetic isotope effect on standard and sample is different. And this is why I do not agree that equal treatment principles apply here and suggest initially that this should've been tested. If one key principle of direct-vapor-equilibration methods - isotopic equilibrium -is violated, there needs to be a solid proof-of-concept in my opinion (and reviewer #1 also raised a similar concern in his initial comments). The revised manuscript does still not clearly address this sufficiently in my opinion. The authors mention the validation standards, but these also don't resolve the abovementioned concerns (because those are also based on exchange with liquid water).

Our standards were based on exchange with wet sand, not liquid water. We tried to make this point clearer in the re-revised manuscript. Nonetheless, we see your point of different matrix potentials potentially having different effects on kinetic isotope fractionation. We admit that this issue needs extra attention prior to our method being field-ready. We also stated this point in the latest version of our manuscript. Thank you for being persistent on this.

In the revised manuscript, the authors also say that the good repeatability of their measurements is a proof of the method; however, a good repeatability has nothing to do with obtaining the correct isotope value. On a side note, and because the authors mention the method: the chance for isotopic equilibrium using the stem borehole method is much higher because no constraining membrane is used and the contact area with the matrix is much greater. Also, equilibrium conditions are tested by determining relative humidity of the sample after passage through the borehole. I simply do not

understand why such a test was never performed with the WIP's (other than Volkmann & Weiler 2014) and is not regularly done when taking vapor samples.

We think that the good precision and accuracy was achieved due to similarity of calibration and validation standards in terms of (among others) probe size, thus vapor source area, and gas flow rate, thus vapor pickup time. Unfortunately, these parameters cannot be unified in the context of the borehole method. For that reason, we denied the applicability of our method for the stem borehole method.

Q: Section 2.1.: This chapter is testing the flow rates through the WIP's and the reader gets the impression those can be chosen arbitrarily; but this is not true. For both soils and xylem, equilibrium fractionation inside of the WIP is required.

A: We argue that identical conditions need to be fulfilled which we did. Our chosen flowrate was identical for all standards and samples. Given the reproducibility of isotope values from nonequilibrium vapor samples in our study, we think that flow rates through identically dimensioned WIPs do not need to facilitate isotope equilibrium. However, this would be an issue requiring extra attention when dealing with e.g. the stem borehole method on trees with heterogeneous diameters. We addressed this issue in the revised discussion.

Additional answer A2: In our study, we tested the necessity of equilibrium for obtaining matrix-bound water isotope data. In a controlled lab experiment, we were able to reproduce the isotopic composition of non-equilibrium based vapor using data from non-equilibrium based standards, which were similar in terms of temperature, vapor concentration and matrix potential of its source. This means, we demonstrated the feasibility of our non-equilibrium-based method under well-defined conditions. We see this as an advancement of the purely equilibrium-based approaches as it may potentially yield additional flexibility in the study design (enabling shorter filling times, higher sample throughput, higher temporal resolution). Regarding the stem borehole method we already denied the applicability of our method due to the inevitable inconsistencies in vapor source areas. Regarding soil vapor sampling, further tests should identify the influence of matrix potential (among others) on non-equilibrium sampling and find a suitable correction.

Q: If flow rates are too high, this requirement cannot be fulfilled anymore. Marshall et al. (2020) provided a way to calculate the maximum flow rate possible with their stem borehole method. However, this seems to not have been tested for WIP's, where the membrane might have an influence on the exchange times. If this was tested, it would be great to cite that or at least provide information on this issue.

A: This matter was tested by Volkmann et al. (2014, doi: 10.5194/hess-18-1819-2014). They optimized the probe dimensions and contact area of the porous tip of the probe to the flowrate demanded by the isotope analyzer. As they aimed at developing an equilibrium-based in situ method, they didn't test other flowrates than the one given by the instrument. We added this information to the revised manuscript. We are aware that we were beyond equilibrium in this study, but aimed at shortening the filling times in the field. We therefore additionally investigated the effect of increasing flowrates and were able to reproduce the intermediate values (validation standards)

Additional answer A2: We are aware of the work of Marshall et al (2020). However, in the context of our study, the concept of a maximum flowrate is not applicable, as we did not aim at equilibrium. Rather, we tested the necessity of equilibrium by producing non-equilibrium-based vapor and reproduce its isotopic composition via established calibration schemes. This turned out to be successful in a controlled lab experiment. Prior to field-readiness, further test will (have to) find ways to correct for the effects of inconsistencies in temperature and matrix potential.